# Hierarchical Overlapping Clustering on Graphs: Cost Function, Algorithm and Scalability

Yicheng Pan [1]  Renjie Chen [1]  Pengyu Long [1]  Bingchen Fan [1]

## Abstract

Hierarchical and overlapping clustering are two prevalent phenomena that often coexist in real-world system. While numerous studies have examined these two structures separately, characterizing and evaluating their hybrid forms remains an open challenge. To bridge this gap, we initiate the study of hierarchical overlapping clustering on graphs by introducing a new cost function and establishing its rationality through several intuitive properties. We further develop an approximation algorithm that achieves a constant approximation factor for its dual version. Our approach employs a recursive overlapping bipartition framework based on local search, enabling a highly scalable speed-up variant. Experimental results demonstrate that this speed-up algorithm outperforms all baseline methods significantly in both effectiveness (across synthetic and real datasets) and scalability.

## 1. Introduction

Clustering on graphs is a major task in machine learning and has a wide range of applications in many areas. Two fundamental categories have attracted in-depth study. The first is hierarchical clustering (HC) that requires a recursive partitioning of a graph into smaller clusters to form a cluster tree (Dasgupta, 2016; Li & Pan, 2016; Cohen-Addad et al., 2019; Charikar & Chatziafratis, 2017; Moseley & Wang, 2017; Naumov et al., 2021). The other is overlapping clustering (OC) that allows data points to belong to multiple clusters (Orecchia et al., 2022; Zhang et al., 2007; Shen et al., 2009; Chen et al., 2010; Nicosia et al., 2009; Whang et al., 2016; Li et al., 2017; Yang & Leskovec, 2012a). Both structural patterns are prevalent in real systems. Compared to using

each structure in isolation, the hybrid structure of hierarchical overlapping clustering (HOC) that permits overlaps between hierarchical clusters offers a more accurate representation of complex relational data observed in real-world scenarios. For instance, in social networks, an individual may belong to multiple distinct groups, and these groups can form larger overlapping communities centered around different themes. In cooperation networks, the coauthors of a paper can be regarded as a base-level cluster, and this cluster may belong to more than one research field due to the diversity of topics. The inherent complexity of the hybrid structure presents a formidable challenge in clustering research. While HC and OC have been extensively studied as separate paradigms, the critical domain of HOC remains largely unexplored. In this paper, we address this problem.

Constructing a cost function is a usual starting point for the research on both HC and OC. Similarly, a proper cost function is helpful to evaluate the quality of HOC, which formulates the HOC problem as an optimization task. In this paper, we propose a new cost function for HOC, and present an approximation algorithm for it. Our contributions are summarized as follows.

(1) **Cost function.** We propose a cost function (Definition 2.8) that is the first one for HOC to our best knowledge. The cost function is evaluated on HOC graphs, and can be unified with Dasgupta's cost function for HC trees in the specific case of non-overlap. We give a comprehensive study on the rationality of this cost function by providing examples and a series of properties including compatibility (Property 2.11), additivity of nodes (Property 2.12) and binary optimality (Property 2.13).

(2) **Approximation algorithm.** Based on our cost function, we formulate the primal and the dual versions of the HOC problem, respectively. We provide an $a = \frac{2}{3\sqrt{6}} - \Theta(\frac{1+\epsilon}{n})$-approximation algorithm (Algorithm 2) for the dual $k$-HOC problem, where $k \in \mathbb{Z}^+$ is an upper bound of key clusters (explained in Definition 2.10), $n$ is the number of graph nodes, and $\epsilon$ is an arbitrarily small positive constant. We also show that our algorithm achieves an approximation factor $(1-a)(1+d_{max}/d_{avg})$ for the primal 2-OC problem.

(3) **Effectiveness and scalability.** We accelerate our ap-

---

[1]State Key Laboratory of Complex & Critical Software Environment, Beihang University. Correspondence to: Yicheng Pan <yichengp@buaa.edu.cn>.

*Proceedings of the 42$^{nd}$ International Conference on Machine Learning*, Vancouver, Canada. PMLR 267, 2025. Copyright 2025 by the author(s).

proximation algorithm by some simple heuristics during the local search process, and verify its effectiveness and scalability by experiments. On effectiveness, experimental results demonstrate that on random graph models with good embedded HOC structures, our algorithm outperforms all other baselines and reconstructs the ground truths almost perfectly. On scalability, benefiting from our elaborate cost function and simple local search strategy, on real datasets with around one million vertices and three million edges, the runtime of our speed-up algorithm implemented on a single laptop is only around $20\%$ of the runtime of the fastest baseline method proposed by (Orecchia et al., 2022) that is run on a server with a cluster of machines.

## 1.1. Related work

**Hierarchical graph clustering.** The most popular cost function for HC is proposed by (Dasgupta, 2016). Given a weighted graph $G = (V, E, w)$ and a cluster tree $T$, Dasgupta's cost is defined as

$$\text{Das}^T(G) = \sum_{i,j \in E} w_{ij} |V(i \vee j)|, \quad (1)$$

where $i \vee j$ denotes the least common ancestor (LCA) of $i$ and $j$ in $T$, $V(i \vee j)$ represents the set of descendant leaf nodes under $i \vee j$, and $|V(i \vee j)|$ is its size. On similarity-based graphs, optimization of HC trees can be performed by minimizing Dasgupta's objective. The intuition is that for a good clustering tree, the edges with large weights ought to be placed as far down from the root as possible, which makes the number of leaves covered by its LCA as small as possible. Dasgupta also showed that both minimizing $\text{Das}^T(G)$ and maximizing $\text{Das}^T(G)$ are NP-hard.

Along this line of study, Dasgupta showed that recursive bipartition applying Arora's seminal algorithm for sparsest cut problem (Arora et al., 2009) yields $O(\log^{1.5} n)$-approximation, and it was improved by (Roy & Pokutta, 2016) and (Charikar & Chatziafratis, 2017; Cohen-Addad et al., 2019) to $O(\log n)$ and $O(\sqrt{\log n})$, respectively. It is also known to be SSE-hard to achieve any constant approximation factor for this objective (Charikar & Chatziafratis, 2017). Moseley and Wang studied the dual of Dasgupta's cost function and showed that the average linkage algorithm achieves a $(1/3)$-approximation (Moseley & Wang, 2017). This factor has been improved by a series of works to 0.336 (Charikar et al., 2019), 0.4246 (Ahmadian et al., 2019) and 0.585 (Alon et al., 2020), respectively. There are also some studies considering the problem of maximizing Dasgupta's cost function on dissimilarity-based graphs (Cohen-Addad et al., 2019; Charikar et al., 2019; Rahgoshay & Salavatipour, 2021; Naumov et al., 2021).

**Overlapping graph clustering.** Newman and Girvan proposed modularity in 2004 (Newman & Girvan, 2004), which

was one of the most popular cost functions for flat non-overlap clustering. Many researchers have extended modularity to the scope of OC. (Nepusz et al., 2008) and (Nicosia et al., 2009) proposed the concept of belonging factor that represents the intensity of a node or an edge belonging to a cluster. A function of the belonging factor was introduced to the definition of modularity to make it applicable to OC, and a heuristic algorithm was proposed based on maximizing the OC modularity. (Zhang et al., 2007), (Shen et al., 2009) and (Chen et al., 2010) also proposed their own definitions of belonging factor and cost functions based on modularity. Inspired by these works, our cost function also utilizes belonging factor for HOC.

On the worst-case guarantee analysis for OC, (Khandekar et al., 2014) formulated the problem of minimizing the maximum or the sum of conductances of overlapping clusters, with or without a bounded number of clusters. They proposed the algorithms that achieve $O(\log n)$-approximation factors for the four kinds of versions, where $n$ is the number of vertices. The techniques behind the proof include the tree decompositions (Räcke, 2002; 2008; Harrelson et al., 2003) and a dynamic programming. As claimed in their work, the complexity of the dynamic program hinders the scalability of their methods.

Another representative work for OC is attributed to (Orecchia et al., 2022), in which two cost functions called $\epsilon$-overlapping ratio-cut ($\epsilon$-ORC) and $\lambda$-hybrid ratio-cut ($\lambda$-HCUT) respectively are proposed for OC with two overlapping clusters. Both cost functions are designed based on the ratio-cut objective, and treat the overlapping part of the two clusters as a penalty. Concretely, given a graph $G = (V, E, w, \mu)$ with non-negative edge weights $w$, vertex measure $\mu$, and two overlapping clusters $L$ and $R$ of vertices, they define two ratio-cut-like measures to be $q_E[L, R] = w(L \setminus R, R \setminus L) / \min\{\mu(L), \mu(R)\}$ and $q_V[L, R] = \mu(L \cap R) / \min\{\mu(L), \mu(R)\}$. Then the $\epsilon$-ORC problem is defined to be the minimization of $q_E[L, R]$ under the condition that $q_V[L, R] \leq \epsilon$, and the $\lambda$-HCUT problem is the minimization of $q_E[L, R] + \lambda q_V[L, R]$. These two problems are defined with hyper-parameters, which restricts the applications and scalability of OC algorithms that solve them. Moreover, since the edge weights $w$ and vertex measure $\mu$ are usually derived from independent systems and have different units, the linear combination of $q_E[L, R]$ and $q_V[L, R]$ in $\lambda$-HCUT is less explainable. However, for both $\epsilon$-ORC and $\lambda$-HCUT, (Orecchia et al., 2022) gave a nearly-linear-time $O(\log n)$-approximation algorithm called $cm + improve$.

With regard to HOC, there is much less work. There are two methods for dissimilarity-based vector data, for which some heuristics based on density criterion (Jeantet et al., 2020) and cut metrics (Gama et al., 2018) are utilized during the

clustering process. However, no cost function and theoretical analysis have been developed yet, which is just what our work addresses.

## 2. A cost function for HOC

In this section, we formulate our cost function for HOC. First of all, we briefly introduce the underlying idea. HOC can be represented by a directed acyclic graph, called HOC graph, that is a natural generalization of HC tree. Inspired by Dasgupta's cost function (Eq. 1) for HC, we extend the LCA of an edge to its minimal common ancestor (MCA) set, and introduce the belonging factor to measure the degree by which a node, a cluster, or an edge belongs to an ancestor. Intuitively, for a similarity-based graph, a quality HOC graph should contain heavy edges into clusters that are small and as far down from the root of the HOC graph as possible. Overlapping is desirable when a node has strong connections to more than one cluster simultaneously, in which case, the belonging factor allows to suppress the cost contributed by the edges incident to that node. This is the crucial idea of our cost function for HOC.

**Preliminaries.** An undirected weighted graph $G = (V, E, w)$ is specified by a node set $V$, an edge set $E \subseteq \{(u,v)|u, v \in V\}$, and a weight function $w : E \to \mathbb{R}^+$. Let $n = |V|$ and $m = |E|$ represent the number of nodes and that of edges, respectively. The degree of a node $u$, denoted by $d_u$, is the sum of weights of all edges incident to $u$, i.e., $d_u = \sum_{(u,v)\in E} w(u,v)$. Denote by $G[U]$ the induced subgraph of $G$ on the node set $U$. For any $A, B \subseteq V$, let $E(A) = \{(u,v)|(u,v) \in E, u, v \in A\}$, $E(A,B) = \{(u,v)|(u,v) \in E, u \in A, v \in B\}$, $w(A) = \sum_{(u,v)\in E(A)} w(u,v)$, $w(A,B) = \sum_{(u,v)\in E(A,B)} w(u,v)$. For a node $v \in V, w(v,A) = \sum_{a\in A,(v,a)\in E} w(v,a)$. For any $E' \subseteq E, w(E') = \sum_{e\in E'} w(e)$.

Partial ordering relationship of two nodes $N$ and $N'$ on a directed acyclic graph $D$, denoted by $N \leq N'$, means that $N'$ is reachable from $N$, and we say that $N$ and $N'$ are *comparable* in this case, and *incomparable* otherwise. An *anti-chain* $L = \{N_1, N_2, N_3, ...\}$ on $D$ is a set of nodes on $D$ satisfying that any two nodes in $L$ are incomparable.

HC on graph $G$ is represented by an HC tree $T$. It has $n$ leaves corresponding to the nodes of $G$. For any internal node $N$, let $V(N)$ be the set of leaves that treat $N$ as an ancestor. Let $u \vee v$ be $u$ and $v$'s LCA on $T$. A weighted graph $G = (V, E, w)$ is *similarity-based* if the larger $w(u,v)$ is, the more similar $u$ and $v$ are. The cost function for HOC discussed in this paper is proposed for similarity-based graphs.

**Definition 2.1** (HOC graph). Given a graph $G$, an HOC $D$ on $G$ is a directed acyclic graph that satisfies the following three constraints: (1) There is only one node of $D$ with out-degree of 0, referred to as the root node and denoted

by $R$. (2) There are $n$ nodes of $D$ with in-degree of 0, corresponding to all the nodes in $V$, referred to as leaf nodes. (3) For each non-root node of $D$, its parent node set $\{N_1, N_2, ...\}$, which is the collection of nodes it points to directly, forms an anti-chain.

On an HOC graph, two nodes satisfying $X \leq Y$ means that $V(X)$ is a subset of $V(Y)$. Note that we do not need the converse also holds, because although syntactically we have $V(X) \subseteq V(Y)$, semantically in practice, $X$ and $Y$ may have unrelated meanings from two different systems that are organized by different mechanisms. HOC graph extends the concept of HC tree by allowing each non-root node to have multiple parent nodes that are incomparable with each other. It is a canonical representation for hierarchical set containment that a subset is only allowed to point to a minimal set that contains it. If each non-root node has out-degree one, an HOC graph degenerates into an HC tree. The *distance* $dis(X, Y)$ is the length of the shortest path from $X$ to $Y$. The *height* of $D$ is the maximum distance from any leaf to the root, that is. $\max_{v\in V} dis(v, R)$. The *width* of an HOC graph is the length of the longest anti-chain that consists of non-leaf nodes. For any node $N$ on $D$, Let $N^-$ denote the set of $N$'s parent nodes and $N_-$ denote the set of $N$'s children nodes. Figure 1 demonstrates three HOC graphs of height 2 for a graph $G$ that consists of two triangles intersecting at a single node, in which $D_2$ is an HC tree without overlap, while the other two are overlapping..

**Definition 2.2** (MCA set). The MCA set for nodes $u$ and $v$ in $D$ is defined as $M_{uv} = \{N|N \in D, u, v \in V(N), \text{ and } \forall X \in N_-, u \notin V(X) \text{ or } v \notin V(X)\}$.

The term "minimal" in the above definition means that any child node of this common ancestor of $u$ and $v$ is not a common ancestor any more, and thus cannot be further reduced. This is an extension of the unique LCA on HC trees to multiple ones on HOC graphs. For convenience, when $u$, $v$ are two endpoints of an edge, we also say a common ancestor of this edge $(u, v)$. As illustrated in $D_3$ of Figure 1, $M_{bc} = \{N_1, N_2\}$, $M_{ae} = \{R\}$.

Then we introduce belonging factor that is a key ingredient of our cost function. We define two kinds of belonging factors on an HOC graph $D$, node-to-node and edge-to-node belonging factors, that are generalizations of those proposed by (Nepusz et al., 2008) and (Nicosia et al., 2009) for OC. The belonging factor of node $X$ (resp. edge $(u, v)$) to node $Y$ represents the degree for which $X$ (resp. edge $(u, v)$) belongs to $Y$.

**Definition 2.3** (node-to-node belonging factor). The node-to-node belonging factor of $X$ to $Y$, denoted by $\alpha_{X,Y}$, is defined recursively. First, define the node-to-node belonging factor for each parent-child node pair on $D$, whose value can be assigned freely as long as it satisfies the following two constraints: (1) $0 \leq \alpha_{X,Y} \leq 1$ for all $X \in D$ and

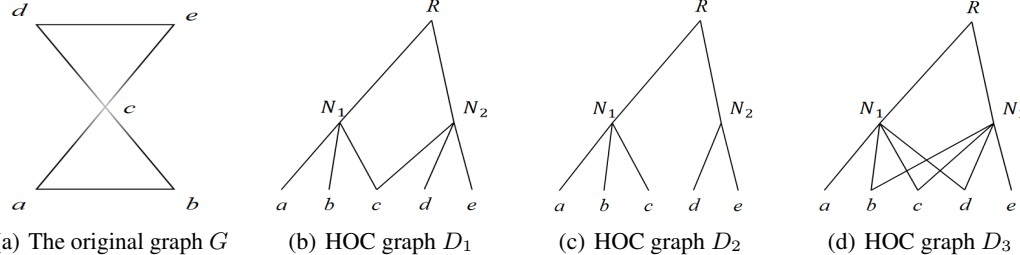

Figure 1. An example of HOC graphs. The costs $H^{D_1}(G) = 18$, $H^{D_2}(G) = 21$, $H^{D_3}(G) = 24$.

$Y \in X^-$; (2) $\sum_{Y \in X^-} \alpha_{X,Y} = 1$ for each non-root node $X \in D$. Second, for other relationships of $X$ and $Y$, $\alpha_{X,Y}$ is defined as

$$\alpha_{X,Y} = \begin{cases} \sum_{N \in X^-} \alpha_{X,N} \cdot \alpha_{N,Y} & X \leq Y, X \neq Y \\ 1 & X = Y \\ 0 & \text{otherwise} \end{cases} \quad (2)$$

To better understand the belonging factor, we give an equivalent and intuitive definition via path-oriented interpretation, thereby uncovering the multiplicative intuition embedded in this concept. For any two comparable nodes $X \leq Y$, denote by $P_{X,Y}$ the set of all paths from $X$ to $Y$. For each path $p = [p_0, p_1, ..., p_{len(p)}] \in P_{X,Y}$, let $p_0 = X$, $p_{len(p)} = Y$, $len(p)$ be the length of $p$. Then the node-to-node belonging factor $\alpha_{X,Y}$ is equivalently defined as $\alpha_{X,Y} = \sum_{p \in P_{X,Y}} \prod_{i=0}^{len(p)-1} \alpha_{p_i, p_{i+1}}$ if $X \leq Y, X \neq Y$, and has the same values as Eq. (2) for the other two cases. That is, $\alpha_{X,Y}$ is the sum of the multiplication of all belonging factors of parent-child pairs along each path from $X$ to $Y$. The equivalence is easy to verify by induction. The node-to-node belonging factor has some fundamental properties.

*Property* 2.4. If $Y$ is the only parent node of $X$, then $\alpha_{X,Y} = 1$.

*Property* 2.5. The node-to-node belonging factor of any node to the root is 1, that is, $\alpha_{N,R} = 1$ for any $N \in D$.

*Property* 2.6. For two nodes $X, Y$ of $D$ satisfying $X \leq Y$, and any node set $S$, if (1) $S$ is an anti-chain, (2) for any $p \in P_{X,Y}, |p \cap S| = 1$, then $\alpha_{X,Y} = \sum_{N \in S} \alpha_{X,N} \cdot \alpha_{N,Y}$.

Property 2.4 unifies HOC graph and common HC tree. Property 2.5 coincides with the common sense that any cluster and leaf belong totally to the root. Property 2.6 implies that the node-to-node belonging factor of $X$ to its ancestor $Y$ can be disassembled by some (minimal) anti-chain that blockades all paths from $X$ to $Y$. The proofs of the above properties are provided in Appendix A.1. Based on node-to-node belonging factor, we define the edge-to-node belonging factor as follows.

**Definition 2.7** (edge-to-node belonging factor). For an edge

$(u, v)$ in graph $G$, let $X \in M_{uv}$ be one of its MCA. The edge-to-node belonging factor $\beta^X_{(u,v)}$ of $(u, v)$ with respect to $X$ is defined as $\beta^X_{(u,v)} = f^X_{(u,v)} / \sum_{Y \in M_{uv}} f^Y_{(u,v)}$, where $f^X_{(u,v)} = \alpha_{u,X} \cdot \alpha_{v,X}$.

$\beta^X_{(u,v)}$ is normalized over all MCAs of $(u, v)$ to guarantee that the mass of its belonging factors to its all MCAs sums up to 1. A natural option is the uniform allocation to each parent. That is, for each non-root node $X \in D$ and $Y \in X^-$,

$$\alpha_{X,Y} = \frac{1}{|X^-|}. \quad (3)$$

We adopt this definition of belonging factor in Section 3. As illustrated in $D_1$ of Figure 1, the leaf $c$ has two minimal ancestors $N_1$ and $N_2$, for each of which has belonging factor $1/2$, and all edges in $G$ belongs totally to $N_1$ or $N_2$. In $D_3$, the leaves $b$, $c$ and $d$ have both $N_1$ and $N_2$ as their minimal ancestors with belonging factor $1/2$ to each, and the edge-to-node belonging factors of $(b, c)$ and $(c, d)$ to either $N_1$ or $N_2$ are $1/2$. We also demonstrate another toy example of 3-length path in Appendix A.2.

Now, we are ready to introduce our HOC cost function based on the edge-to-node belonging factor.

**Definition 2.8** (cost function for HOC). Given a graph $G$ and an HOC graph $D$ of $G$, the cost function of $D$ on $G$ is defined as

$$H^D(G) = \sum_{(u,v) \in E} \left( w(u,v) \cdot \sum_{N \in M_{uv}} \beta^N_{(u,v)} \cdot |V(N)| \right).$$

The cost function contains two summations. The first is over all edges, and the second is over the MCAs of the endpoints of corresponding edge. Compared with Dasgupta's cost function (1), $H^D(G)$ generalizes it from HC to HOC by assigning a belonging factor for each MCA of each edge.

**Definition 2.9** (HOC problem). The HOC problem on a similarity-based graph $G$ is defined as $\min_D H^D(G)$ under some proper constraints on the HOC graph $D$.

The intuition behind minimizing the cost function on similarity-based graphs is the same as Dasgupta's cost function, that is, to assign heavy edges to the clusters as small as possible. On an HOC graph, this can be achieved by ensuring that the MCAs of these edges are as far down from the root as possible.

As illustrated in Figure 1, according to Definition 2.8, the costs of $D_1$, $D_2$ and $D_3$ are 18, 21 and 24, respectively. We provide the calculating process in Appendix A.3. We can see that $D_1$, which is intuitively more reasonable than $D_2$ and $D_3$, has the smallest cost. It is also demonstrated that introducing overlaps has the advantage of reducing the MCAs of edges, thereby decreasing their costs (compare $D_2$ to $D_1$). On the other hand, this comes at the expense of increasing the number of descendant leaves of the ancestors. So, excessive overlap gets punished (compare $D_1$ to $D_3$). Therefore, our cost function balances insufficient and excessive overlaps.

**Remark.** Note that HOC is quite different from HC since it allows possibly an exponential number of overlapping clusters without any restriction, and thus proper constraints on $D$ are necessary. However, we need to be very careful in formulating the constraints. In fact, there is a trivial solution that treats two endpoints of each edge as a cluster, which achieves the minimum cost $2w(E)$. This solution is in fact an intuitive way for overlapping cluster settings since only similar nodes can be linked together, but due to the large number of clusters, it makes no sense. This is quite different from the optimization of Dasgupta's cost function for HC. A natural restriction on HOC graphs may be on the number of clusters. However, since an HOC graph has hierarchical clusters, we seek to have a meaningful constraint on the cluster number. To this end, we utilize the width of an HOC graph, which is the length of the longest anti-chain on it.

**Definition 2.10** ($k$-HOC problem)**.** The $k$-HOC problem on a similarity-based graph $G$ is defined as $\min_D H^D(G)$ for which the width of the HOC graph $D$ is at most $k$.

To better understand this problem, let us consider a non-overlapping HC tree first. Therein, the width means the largest number of bottom and smallest non-overlapping clusters whose children are all leaves. They can be considered as a set of key clusters that are closest to the leaves on the tree. Similarly, on an HOC graph, since a longest anti-chain is located, intuitively, as far down from the root as possible to blockade leaf-to-root paths, the width measures the number of incomparable bottom clusters that contain the leaves.

Moreover, we define $k$-*OC problem* to be the $k$-HOC problem in which we additionally restrict the height of $D$ at most 2, in which case HOC degrades to OC. A fundamental case is 2-OC that allows only two overlapping clusters. 2-OC can be considered as a key ingredient of HOC with multiple clusters since it could be a nice way to construct a $k$-HOC

graph by recursively calling 2-OC algorithm in a top-down fashion. In Section 3, our algorithm for $k$-HOC proceeds in this way.

Next, we give some fundamental properties of our HOC cost function, and prove them in Appendix A.4.

*Property* 2.11 (Compatibility)**.** If $D$ is an HC tree, then $H^D(G) = \mathrm{Das}^D(G) = \sum_{(u,v)\in E} w(u,v) \cdot |u \vee v|$.

*Property* 2.12 (Additivity on nodes)**.** For any node $N$ on $D$, let $E_N^D$ denote the set of edges treating $N$ as an MCA, i.e., $E_N^D = \{(u,v)|(u,v) \in E, N \in M_{uv}\}$. The HOC cost function can be rewritten as $H^D(G) = \sum_{N\in D} \left( |V(N)| \cdot \sum_{(u,v)\in E_N^D} \beta_{(u,v)}^N \cdot w(u,v) \right)$.

*Property* 2.13 (Binary optimality)**.** As long as the constraints for the HOC problem are not violated, binary branching on each non-leaf node (i.e., the number of children is at most 2) is desirable for an optimal HOC graph.

Property 2.11 indicates that our cost function for HOC can be unified with Dasgupta's cost. That is, under the constraint of hierarchical non-overlapping clustering, our cost function for HOC problem degrades to Dasgupta's objective whose optimization is NP-hard (Dasgupta, 2016). Property 2.12 provides an alternative interpretation of the cost function from another perspective, for which it can be seen as the sum of costs associated with each node. Note that Dasgupta's cost also has the last two properties.

**Primal and dual versions of HOC problem.** Next, we introduce the primal and the dual HOC problems. Note that besides the trivial lower bound $2w(E)$ for $H^D(G)$, we also have a trivial upper bound $nw(E)$, since the size of any common ancestor of two leaves on $D$ is at most $n$. So, we define the *primal $k$-HOC problem*, denoted by $k$-HOC-P, to be $\min_D\{H^D(G)\}$ as Definition 2.9. We define the *dual $k$-HOC problem*, denoted by $k$-HOC-D, to be $\max_D\{nw(E) - H^D(G)\}$, that is, by Definition 2.8, maximizing $\sum_{(u,v)\in E} \left( w(u,v) \cdot \sum_{N\in M_{uv}} \beta_{(u,v)}^N (n - |V(N)|) \right)$ over $D$. This cost is the compatible counterpart of Mosley and Wang's dual HC cost (Moseley & Wang, 2017) in the HOC setting. The solutions to the primal and dual problems achieve optima on the same HOC graph. Similarly, let $k$-OC-P and $k$-OC-D denote the corresponding versions of OC problem, respectively.

## 3. An algorithm for $k$-HOC

In this section, we propose our algorithm for the $k$-HOC problem. We adopt Eq. (3) as the node-to-node belonging factor. Since our algorithm is a recursive process of 2-OC, we study 2-OC first, and then apply it to $k$-HOC. Note that overlapping bipartition is a fundamental problem for OC, and is also the theme (Orecchia et al., 2022)'s theoretical

---

**Algorithm 1** Algorithm for 2-OC

   **Input:** an undirected graph $G = (V, E, w)$
   **Output:** node sets $A$, $B$ and $C$ for 2-OC
   $n \leftarrow |V|, p \leftarrow \frac{1}{\sqrt{6}}, x \leftarrow \frac{p}{1+2p}$
   Define a new $cost_{temp}(A, B, C) = w(E) - w(A, C) + xw(B)$
   Divide arbitrarily $V$ into three disjoint parts $A$, $B$, $C$ satisfying $|A| = |C| = pn$, $|B| = (1 - 2p)n$, such that the edge with the maximum weight are not in $E(A, C)$
   **repeat**
      Exchange any two nodes from different sets of $A$, $B$, $C$ whenever $cost_{temp}$ can be amplified by more than $1 + \epsilon/n^2$ times
   **until** get stuck
   return $A$, $B$, $C$.

---

study. So our 2-OC problem has its own interests from the perspective of our new cost function. Compared with the complicated "cut-matching and improve" approach of (Orecchia et al., 2022), our method for 2-OC takes a simple local search heuristic based on our cost function, which makes our algorithm much more scalable.

### 3.1. An approximation algorithm for 2-OC

**Cost functions for 2-OC.** In the 2-OC setting, given a graph $G = (V, E, w)$, we restrict the height of the HOC graph to 2 and the number of children of the root $R$ to 2. Suppose that two clusters $N_1 = A \cup B$ and $N_2 = C \cup B$ overlap on $B$. By Definition 2.8, the cost function of 2-OC-P can be formulated as $cost_{primal}(A, B, C) = [w(A) + w(A, B)](|A| + |B|) + [w(B, C) + w(C)](|B| + |C|) + \frac{(|A| + 2|B| + |C|)w(B)}{2} + w(A, C)n$, and 2-OC-P can be formulated as

$$\min_{A, B, C \subseteq V} cost_{primal}(A, B, C) \qquad \text{(2-OC-P)}$$

We also have the cost function $cost_{dual}(A, B, C) = (w(A + B) - \frac{w(B)}{2})|C| + (w(B + C) - \frac{w(B)}{2})|A|$ for 2-OC-D, and 2-OC-D can be formulated as

$$\max_{A, B, C \subseteq V} cost_{dual}(A, B, C) \qquad \text{(2-OC-D)}$$

The derivation of the forms of $cost_{primal}$ and $cost_{dual}$ is presented in Appendix B.1. We remark that although our cost functions for 2-OC look complicated, they are hyper-parameter free and natural from the perspective of HOC, which is superior to the objective proposed by (Orecchia et al., 2022). Then we propose our algorithm for 2-OC.

**Approximation algorithm for 2-OC.** Algorithm 1 is a simple local search process for 2-OC. It first defines a surrogate cost function $cost_{temp}(A, B, C) = w(E) - w(A, C) + xw(B)$, and initializes $A$, $B$, $C$ arbitrarily (e.g. a random

initialization). After that, the nodes in $A$, $B$, $C$ exchange pairwisely on the condition that current cost can be amplified by $1 + \epsilon/n^2$ times, that is, $cost_{temp}(A', B', C') > (1+\epsilon/n^2)cost_{temp}(A, B, C)$, where $A'$, $B'$, $C'$ are the node sets after exchanging corresponding to $A$, $B$, $C$ respectively, and $\epsilon > 0$ is a small constant. It doesn't terminate until no pair of nodes meets the exchange condition. For the worst-case guarantee, we have the following theorem.

**Theorem 3.1.** *Algorithm 1 achieves an approximation factor $a = \frac{2}{3\sqrt{6}} - \Theta(\frac{1+\epsilon}{n})$ for 2-OC-D with time complexity $O(\epsilon^{-1}n^4 \log m)$ for any $\epsilon > 0$.*

The idea of the proof of Theorem 3.1 is as follows. Since $nw(E)$ is a trivial upper bound on the objective function, we only have to show that $cost_{dual} \geq \left(\frac{2}{3\sqrt{6}} - \Theta(\frac{1+\epsilon}{n})\right) \cdot nw(E)$. Since Algorithm 1 fixes the sizes of $A$, $B$, $C$, we only need to build the relationship between $w(E)$ and edge weights of different parts in the cost function. A lower bound on the latter related to $w(E)$ (Inequality (11)) can be obtained by the three stuck exchange conditions when the iteration terminates. The detailed proof of Theorem 3.1 is provided in Appendix B.2. Moreover, we have the following proposition to demonstrate the tightness of our guarantee in some sense.

**Proposition 3.2.** *For the complete graph $K_n$, the optimal 2-OC-D value $OPT(K_n) = \left(\frac{2}{3\sqrt{6}} - \Theta(\frac{1}{n})\right)nw(E)$.*

Proposition 3.2 implies that, if an approximation algorithm for 2-OC-D is designed based on the upper bound $nw(E)$ of $cost_{dual}$, the optimal approximation factor cannot be better than $\frac{2}{3\sqrt{6}} - \Theta(\frac{1}{n})$. Its proof is provided in Appendix B.3.

**Approximation guarantee algorithm for 2-OC-P.** We also show that Algorithm 1 is actually a good approximation algorithm for 2-OC-P. We simply treat the output of Algorithm 1 as the result for 2-OC-P, and we have the following approximation guarantee for 2-OC-P.

**Theorem 3.3.** *The approximation factor of Algorithm 1 for 2-OC-P is $(1 - a)(1 + d_{max}/d_{avg})$, where $d_{max}$ is the maximum degree of all nodes, $d_{avg}$ is the average degree, and $a = \frac{2}{3\sqrt{6}} - \Theta(\frac{1+\epsilon}{n})$.*

We prove Theorem 3.3 in Appendix B.4. By this theorem, we have two corollaries for regular and bounded-degree graphs, respectively.

**Corollary 3.4.** *If the graph $G$ is d-regular, then the approximation factor of Algorithm 1 for 2-OC-P on $G$ is $2(1 - a)$.*

**Corollary 3.5.** *If the degree of each node in unweighted graph $G$ is upper bounded by a constant $d$, then the approximation factor of Algorithm 1 for 2-OC-P on $G$ is $(1 + d)(1 - a)$.*

---

**Algorithm 2** Algorithm for $k$-HOC

---

**Input:** an undirected graph $G = (V, E, w)$, an integer $k \leq n$
**Output:** a $k$-HOC graph $D$
initialize $D$ with all leaves pointing to the root $r$
$\mathcal{S} \leftarrow \{r\}$
**repeat**
  $X_{max} \leftarrow \arg\max_{X \in \mathcal{S}}\{\Delta(X)\}$
  Apply Algorithm 1 to the subgraph induced by $X_{max}$ and obtain two internal nodes $X_L, X_R$
  $\mathcal{S} \leftarrow \mathcal{S} \setminus \{X_{max}\}$
  $\mathcal{S} \leftarrow \mathcal{S} \cup \{X_L, X_R\}$
  Add $X_L$ and $X_R$ to $D$ as $X_{max}$'s left and right child, respectively, and redirect the leaves to their corresponding parents
**until** $|\mathcal{S}| = k$ or each set in $\mathcal{S}$ has a size 2
Merge identical nodes in $D$ into a single one while keeping all the connections on them
Remove all redundant directed edges $(X, Y)$ for which there is another path from $X$ to $Y$ in $D$
return $\mathcal{D}$.

---

### 3.2. An approximation algorithm for $k$-HOC

Now we turn to $k$-HOC. We assume that $k \leq n$ for practical significance. Since the width of the HOC graph is no more than $k$, we invoke the 2-OC algorithm $k - 1$ times to guarantee this. We first construct a binary tree (excluding the leaves) for the internal nodes, and then merge the identical ones that consists of the same set of leaves, while keeping all directed edges on them. In each iteration, the splitting cluster is chosen greedily according to the relative benefit of cost. Formally, we define $\Delta(X) = \frac{cost_{dual}(X)}{|V(X)|w(X)}$ for the most bottom clusters $X$, where $cost_{dual}(X)$ is the dual cost obtained by the 2-OC algorithm on the subgraph induced by $X$. In each round, we choose the $X$ with the largest $\Delta(X)$ to split. This procedure is described as Algorithm 2.

Now we show that $D$ output by Algorithm 2 is a legal $k$-HOC graph. By definition 2.1, we have to show that the parents of any non-root node form an anti-chain, and the width of $D$ is at most $k$. For any node $X$, since we remove all the directed edges $(X, Y)$ for which there is another path from $X$ to $Y$ in $D$, $X^-$ is obviously an anti-chain. Since the 2-OC algorithm is called for at most $k - 1$ times, the width of $D$ before merging is at most $k$. Since merging does not increase the width, the final $D$ is a $k$-HOC graph.

**Time complexity.** The runtime of Algorithm 2 consists of three parts: the recursive division, merging identical nodes and removing redundant edges. In the division step, it calls Algorithm 1 at most $k - 1$ times, which takes $O(k\epsilon^{-1}n^4 \log m)$ time. In the node merging step, an efficient implementation leverages bitmaps and sorting. The

bitmap of each internal node whose length is $n$ indicates the membership of each leaf. It is necessary to check whether $O(k)$ bitmaps are the same, which takes $O(nk \log k)$ time. In the edge removing step, a redundant edge $(X, Y)$ can be identified by reversing it and checking the reachability from $X$ to $Y$. This takes $O((n + k)^2)$ time. Combining all three parts, the total runtime is $O(k\epsilon^{-1}n^4 \log m)$.

For the approximation guarantee, we have

**Theorem 3.6.** *Algorithm 2 achieves an approximation factor* $\frac{2}{3\sqrt{6}} - \Theta(\frac{1+\epsilon}{n})$ *for the dual $k$-HOC problem.*

*Proof.* To prove Theorem 3.6, we only have to show that the dual cost is at least that of Algorithm 1 which is lower bounded by $\left(\frac{2}{3\sqrt{6}} - \Theta(\frac{1+\epsilon}{n})\right) \cdot nw(E)$. Then the approximation factor follows from the fact that $nw(E)$ upper bounds the dual cost of any HOC graph.

Note that Algorithm 1 achieves a dual cost at least $\left(\frac{2}{3\sqrt{6}} - \Theta(\frac{1+\epsilon}{n})\right) \cdot nw(E)$ for 2-OC-D, in which $nw(E)$ is an upper bound for the cost of the dual HOC problem with any constraints. So, Theorem 3.6 follows if the final dual cost is no less than the one after the first round of invoking Algorithm 1.

Let $N_1$ and $N_2$ be the two overlapping clusters that Algorithm 1 yields in the first round of the repeat loop, and $e \in E$ be an edge that treats $N_1$ or $N_2$ as a common ancestor. Then the root $r$ is not an MCA of $e$, since otherwise, $r$ forms an chain with $N_1$ or $N_2$. In the next iterations, $r$ will not be included in the MCA set of $e$ during both splitting and merging process. Since $N_1$ and $N_2$ have the same size, no matter how the belonging factors of $e$ change, the final primal cost that $e$ contributes will not exceed that after the first round of invoking Algorithm 1. This proves Theorem 3.6. $\square$

**A speed-up version.** Algorithms 1 and 2 have theoretical significance, but are not efficient enough in practice. Moreover, the setting of fixed sizes of $A$, $B$ and $C$ in Algorithms 1 is too rigid to fit for flexible scenarios. For scalability and practical application of our algorithm, we propose the speed-up version (Algorithm 3) of Algorithm 1 and use it in Algorithm 2 to yield our speed-up algorithm for $k$-HOC. Their effectiveness and scalability will be verified in Section 4. Two easy heuristics are proposed for acceleration, and the pseudocode is presented as Algorithm 3.

(1) Initialization based on ratio-cut (Hagen & Kahng, 1992): Instead of the random strategy for the initial trisection, we use the spectral clustering algorithm RatioCut to split the node set into two pieces, denoted by $X, Y$, let $A = X$, $B = \emptyset$, $C = Y$. Then the nodes move greedily among $A$, $B$, and $C$ instead of exchange.

(2) Batch migration: Starting from the initial $A$, $B$, $C$,

---

**Algorithm 3** Speed-up algorithm for 2-OC

---

**Input:** an undirected graph $G = (V, E, w)$, batch migration number $\ell$
**Output:** node sets $A$, $B$ and $C$ for 2-OC
$X, Y \leftarrow RatioCut(G)$
$A \leftarrow X, B \leftarrow \emptyset, C \leftarrow Y$
**repeat**
    Calculate the delta of $cost_{dual}$ when each node moves to the other two sets, and select the one with the larger increment as the potential action at that node
    Let $S$ be the node set that brings $cost_{dual}$ increment
    $t \leftarrow \min\{|S|, \ell\}$
    Move the top-$t$ nodes with the largest increment
**until** $S$ is empty
return $A$, $B$, $C$.

---

calculate the variation of the cost for each node when it moves to another set, and select a batch of $\ell$ nodes with positive and the largest variation of cost to move in one step UNTIL all nodes get stuck. In our experiment, we set $\ell = 32$. If there are less than $\ell$ nodes that need to move, we move all of them.

Replacing Algorithm 1 with Algorithm 3 in Algorithm 2, we get the speed-up version of $k$-HOC algorithm.

## 4. Experiments

In this section, we verify by experiments the effectiveness and scalability of the speed-up version of Algorithm 2, which demonstrates the validity of our cost function as well. Our experiments were performed on a personal computer equipped with a 2.3GHz quad-core Intel i5 processor with 8GB memory. For the source codes and datasets, please refer to `https://github.com/Hardict/HOC`.

**Baselines.** We include three baselines, named Dasgupta, OHC'20 (Jeantet et al., 2020) and $cm + improve$ (Orecchia et al., 2022), respectively. The latter two are not fit for the HOC settings directly and require some modifications. Then we introduce each method and our modifications.

Dasgupta is simply the speed-up version of Algorithm 2 except that it always keeps $B$ empty in the subroutine Algorithm 1. So, it degrades to optimizing Dasgupta's cost with width $k$ in our local search fashion. We adopt this baseline in order to see whether we can really benefit from overlapping compared with the non-overlapping counterpart of Dasgupta's cost.

OHC'20 (Jeantet et al., 2020) works for HOC on dissimilarity-based vector data only. It is a density-based algorithm that proceeds in an agglomerative bottom-up fashion. To fit to similarity-based graph clustering in our settings, we feed to OHC'20 as input the spectral embedding

that consists of the top-$k$ eigenvectors of the Laplacian matrix. Since this method need to deal with all-pair distance, it is not able to work on large graphs.

$cm + improve$ (Orecchia et al., 2022) is a nearly linear-time $O(\log n)$-approximation algorithm that works for overlapping bipartition only. We substitute $cm + improve$ for the 2-OC algorithm that is used in our $k$-HOC process. However, there are many hyper-parameters to adjust in it, and we have chosen some mild values in our experiments.

**Datasets and evaluation.** For synthetic datasets, we employ the overlapping stochastic block model (OSBM), a generalized version of SBM that allows for overlapping clusters. We introduce it in Appendix C.1. In our experiments, we modify OSBM to preset two hierarchies. We assume that each node belongs to at most two clusters. Thus, the total number of nodes is exactly the sum of entries in the upper triangle of the membership matrix $Z$. For simplicity of implementation, all clusters are of the same size and have the same size of overlaps between clusters. For a 2-level hierarchical structure, we choose three probability values $0 \leq p_1 \leq p_2 \leq p_3 \leq 1$, in which $p_1$ is the inter-link probability between clusters on the first level, $p_2$ is the inter-link probability between clusters on the second level, and $p_3$ is the intra-link probability within each cluster. So now, the probability of edge presence between any two nodes in the overlapping region is $1 - (1 - p_3)^2$.

We use NMI for OC (McDaid et al., 2011) for evaluations, and its formal definition is provided in Appendix C.2. Since NMI is only suitable for non-hierarchical clusters, we evaluate our algorithm at each level of HOC graphs. For real datasets, we adopt Amazon, Youtube, and DBLP (Yang & Leskovec, 2012b) that are provided by `http://snap.stanford.edu/data` and are also used in (Orecchia et al., 2022). Due to lacking ground truth for HOC, we onlu evaluate scalability on the real datasets.

**Effectiveness.** We demonstrate in Figure 2 the results on OSBM datasets with varying sizes. We show the time, cost, and NMI results of our $k$-HOC algorithm and the baselines. It can be observed that the runtime of our HOC algorithm that generates four overlapping bottom clusters for dense graph of size 5000 is only around 80s, and that for sparse graph of size $10^5$ is less than 15min, while keeping extremely high NMIs. We do not show the results of OHC'20 for sparse graphs since it is not able to terminate in one hour for a graph of size $10^4$. The cost results indicate that our algorithm outperforms OHC'20, and we have indeed gained benefits of cost from overlapping when compared with the non-overlapping counterpart of Dasgupta's cost. We evaluate NMI on the two hierarchies respectively. For OHC'20, since it cannot restrict the hierarchy numbers, in each round of evaluation, we choose the level that achieves the highest NMI compared with the ground truth. The NMIs of our

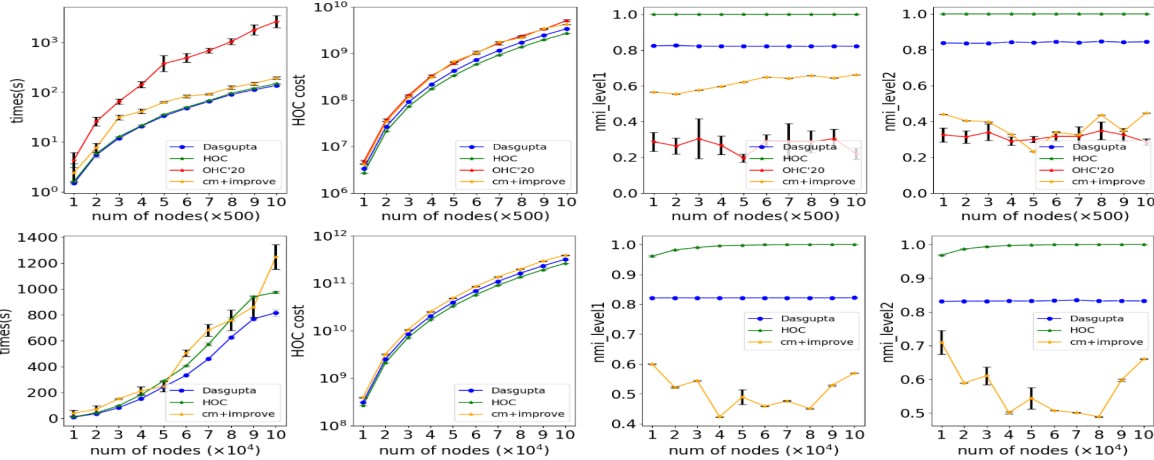

*Figure 2.* The results of time, cost, and NMI. In each figure, the $x$-axis indicates the graph size. The graphs in the first row are small and dense, while those in the second row are large and sparse. We take $k = 4$ in OSBM, and in each cluster, the size ratio of overlapping to non-overlapping is 9:1. In the fist row, $p_1 = 10^{-3}$, $p_2 = 5 \times 10^{-3}$, $p_3 = 0.5$. In the second row, $p_1 = 10^{-4}$, $p_2 = 2 \times 10^{-4}$, $p_3 = 5 \times 10^{-3}$. Regard to the last two columns of NMI results, "level 1" is the first level that contains the two high-level clusters, and "level 2" is the second one that contains the four low-level clusters. Each point is calculated on average over 5 trials, and error bar indicates standard deviation.

$k$-HOC algorithm on both levels are almost 1.0 on dense graphs, and also above 0.95 on sparse graphs, which demonstrates that it achieves high accuracy in reconstruction on each level. We visualize a result in Appendix C.4.

**Scalability.** The first column of Figure 2 has demonstrated that our $k$-HOC algorithm is efficient on synthetic graphs, especially much better than OHC'20. Next, on large real datasets, we show in Table 1 the advantage of our algorithm over $cm + improve$ in scalability. For a fair comparison, we compare our 2-OC algorithm with it. It can be seen that the runtime of our algorithm that is run on a laptop on all datasets is much shorter than the runtime of $cm + improve$ that is run on a server [1]. In particular, on Youtube dataset that has around one million nodes and three million edges, the runtime of our algorithm implemented on a single personal computer is less than 12 minutes, which is only around 20% of the runtime of $cm + improve$ that is run on a server. Although (Orecchia et al., 2022) has showed that $cm + improve$ has nearly linear runtime based on the solid nearly linear-time algorithm of (Chen et al., 2022) for the maximum-flow problem, they actually use the HIPR implementation (Cherkassky et al., 1994) with the push-labeled method. This possibly limits the efficiency of $cm + improve$. The advantage of our algorithm in efficiency

benefits from our elaborate cost function and the simple local search strategy.

*Table 1.* Scalability performance on real datasets

| dataset | n | m | time | cm time |
|---------|------|------|------|---------|
| Amazon | 334863 | 925872 | <3min | 15-18min |
| Youtube | 1134890 | 2987624 | <12min | 55-75min |
| DBLP-all | 317080 | 1049866 | <3min | – |
| DBLP-cm | 83114 | 409541 | <21s | 2-4min |

## 5. Conclusions and future work

**Conclusions.** In this paper, we study the HOC problem on graphs from the aspects of cost function, algorithm and scalability. We have proposed a cost function and given some fundamental properties. We have developed an approximation algorithm that achieves a constant factor for the dual $k$-HOC problem. A speed-up version of our algorithm based on some easy heuristics during local search has good performances in HOC reconstruction and good scalability.

**Future work.** There are many directions worth further study. The first is about approximation algorithm for the primal $k$-HOC problem for $k > 2$. Although we know the complementary relationship between the primal and the dual problems, the approximation guarantees are quite different. The second is about variant versions of the HOC problem, e.g., having other constraints on HOC graphs and alternative definitions of node-to-node and edge-to-node belonging factors. These flexible settings may adapt to different application scenarios.

---

[1]$cm + improve$ can't terminate in a few hours on too large datasets when it is run on our computer. Due to the mismatch of computing resources, the results in the last column of Table 1 are from Table 3 of the original paper (Orecchia et al., 2022) whose experimental operating environment is reported to include a cluster of machines with 24 Cores (2x 24 core Intel Xeon Silver 4116 CPU @ 2.10GHz), 48 threads and 128GB RAM. In a sharp contrast, we have only used a personal computer.

# Acknowledgements

The authors gratefully acknowledge the anonymous reviewers for the constructive comments that have improved this work. This work is partly supported by National Key R&D Program of China (2021YFB3500700), and partly supported by State Key Laboratory of Complex & Critical Software Environment (CCSE-2024ZX-20).

# Impact Statement

This paper presents work whose goal is to advance the field of Machine Learning. There are many potential societal consequences of our work, none which we feel must be specifically highlighted here.

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

# A. Supplements to the cost function for HOC

In this section, we provide some supplementary proofs and examples for our HOC cost function.

## A.1. Proofs of the properties of belonging factor

(1) Proof of Property 2.4

*Proof.* By Definition 2.3, $\sum_{Y \in X^-} \alpha_{X,Y} = 1$. When $Y$ is the only parent of $X$, $\alpha_{X,Y} = 1$. $\square$

(2) Proof of Property 2.5

*Proof.* We prove it by induction. We group the nodes on an HOC graph $D$ by the distances from the root $R$. For each integer $i$, let $S_i = \{N \in D | dis(N, R) = i\}$. Then we prove the property by induction on $i$. For each $N \in S_1$, since $|N^-| = 1$, we have $\alpha_{N,R} = 1$. Suppose that for each $X \in S_k$, $\alpha_{X,R} = 1$, and then for each $N \in S_{k+1}$, by the recursive definition of $\alpha$, $\alpha_{N,R} = \sum_{X \in N^-} \alpha_{N,X} \cdot \alpha_{X,R} = \sum_{X \in N^-} \alpha_{N,X} = 1$. $\square$

(3) Proof of Property 2.6

*Proof.* Since $S$ is an anti-chain and every path from $X$ to $Y$ passes through one node in $S$, all paths from $X$ to $Y$ can be divided into $|S|$ subsets according to whether a node $N \in S$ is on the path or not. Then Property 2.6 can be verified directly by the definition of $\alpha$.

$$
\begin{aligned}
\alpha_{X,Y} &= \sum_{p \in P_{X,Y}} \prod_{i=0}^{len(p)-1} \alpha_{p_i, p_{i+1}} \\
&= \sum_{N \in S} \sum_{p:N \in p} \prod_{i=0}^{len(p)-1} \alpha_{p_i, p_{i+1}} \\
&= \sum_{N \in S} \left( \sum_{p \in P_{X,N}} \prod_{i=0}^{len(p)-1} \alpha_{p_i, p_{i+1}} \right) \left( \sum_{p \in P_{N,Y}} \prod_{i=0}^{len(p)-1} \alpha_{p_i, p_{i+1}} \right) \\
&= \sum_{N \in S} \alpha_{X,N} \cdot \alpha_{N,Y}
\end{aligned}
$$

$\square$

## A.2. A toy example of belonging factor

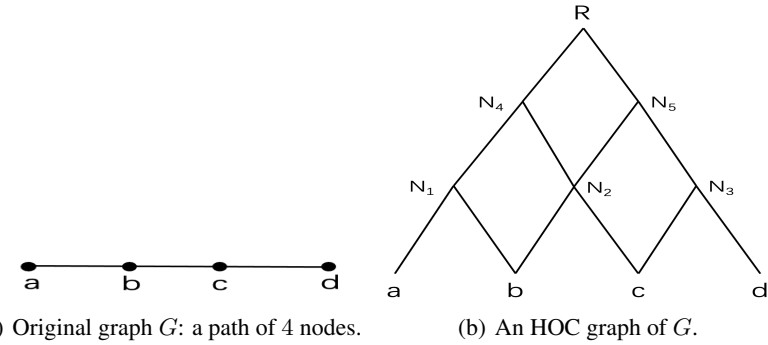

(a) Original graph $G$: a path of $4$ nodes.  (b) An HOC graph of $G$.

*Figure 3.* Illustration of HOC.

In order to better understand node-to-node and edge-to-node belonging factors, we give an example in this section.

As shown in Figure 3, graph $G$ is a path of 4 nodes, and a possible HOC graph is shown in Figure (b).

Table 2 demonstrates the MCA set of each leaf node pair. Table 3 shows the node-to-node belonging factor of each child-to-parent node pair on the HOC graph, and those of any others can be calculated by Definition 2.3. For example,

$$\alpha_{b,N_4} = \alpha_{b,N_1} \cdot \alpha_{N_1,N_4} + \alpha_{b,N_2} \cdot \alpha_{N_2,N_4} = \frac{1}{2} \times 1 + \frac{1}{2} \times \frac{1}{2} = \frac{3}{4},$$

$$\alpha_{b,N_5} = \alpha_{b,N_2} \cdot \alpha_{N_2,N_5} = \frac{1}{2} \times \frac{1}{2} = \frac{1}{4}.$$

Table 2. MCA sets

| node pair $(u,v)$ | $(a,b)$ | $(a,c)$ | $(a,d)$ | $(b,c)$ | $(b,d)$ | $(c,d)$ |
|---|---|---|---|---|---|---|
| $M_{uv}$ | $\{N_1\}$ | $\{N_2\}$ | $\{R\}$ | $\{N_2\}$ | $\{N_5\}$ | $\{N_3\}$ |

Table 3. node-to-node belonging factors (child to parents)

| node pair $(X,Y)$ | $(a,N_1)$ | $(b,N_1)$ | $(b,N_2)$ | $(c,N_2)$ | $(c,N_3)$ | $(d,N_3)$ |
|---|---|---|---|---|---|---|
| $\alpha_{X,Y}$ | 1 | 1/2 | 1/2 | 1/2 | 1/2 | 1 |
| node pair $(X,Y)$ | $(N_1,N_4)$ | $(N_2,N_4)$ | $(N_2,N_5)$ | $(N_3,N_5)$ | $(N_4,R)$ | $(N_5,R)$ |
| $\alpha_{X,Y}$ | 1 | 1/2 | 1/2 | 1 | 1 | 1 |

We can also verify the properties of the node-to-node belonging factor. Here we only verify Property 2.6, and other properties can be easily verified. Let $X = b$, $Y = N_4$, $S = \{N_1, N_2\}$. We can verify that $S$ satisfies all conditions of Property 2.6. Then

$$\alpha_{b,N_4} = \sum_{N \in S} \alpha_{b,N} \cdot \alpha_{N,N_4} = \alpha_{b,N_1} \cdot \alpha_{N_1,N_4} + \alpha_{b,N_2} \cdot \alpha_{N_2,N_4} = \frac{1}{2} \times 1 + \frac{1}{2} \times \frac{1}{2} = \frac{3}{4}.$$

Again, let $X = b$, $Y = R$, $S = \{N_4, N_5\}$. Then

$$\alpha_{b,R} = \sum_{N \in S} \alpha_{b,N} \cdot \alpha_{N,R} = \alpha_{b,N_4} \cdot \alpha_{N_4,R} + \alpha_{b,N_5} \cdot \alpha_{N_5,R} = \frac{3}{4} \times 1 + \frac{1}{4} \times 1 = 1.$$

Table 4. edge-to-node belonging factors

| edge $(u,v)$ | MCAs | $\beta^N_{(u,v)}$ |
|---|---|---|
| $(a,b)$ | $N_1$ | 1 |
| $(b,c)$ | $N_2$ | 1 |
| $(c,d)$ | $N_3$ | 1 |

Table 4 shows edge-to-node belonging factors of all edges and their MCAs. Because each edge has only one MCA, its edge-to-node belonging factor is 1.

### A.3. Cost calculation for the running example

For reading convenience, we demonstrate the example again in Figure 4.

In $D_1$, all edges have only one MCA, so the edge-to-node belonging factors of them are 1. The graph contains 6 edges in all, and each MCA has 3 descendant leaf nodes, resulting in the cost $H^{D_1}(G) = 6 \times 3 = 18$.

$D_2$ is not overlapping. For $(a,b)$, $(a,c)$, $(b,c)$, their MCA has 3 descendant leaf nodes. For $(d,e)$, the MCA has 2 descendant leaf nodes. For $(c,d)$, $(c,e)$, their MCA has 5 descendant leaf nodes. All together, the cost $H^{D_2}(G) = 3 \times 3 + 2 + 2 \times 5 = 21$.

In $D_3$, consider 6 terms separately corresponding to the 6 edges. Taking $(b,c)$ as an example, it has two MCAs. Due to symmetry, the edge-to-node belonging factors of $(b,c)$ regarding to both ancestors are 0.5. Therefore, the cost contributed by $(b,c)$ is $0.5 \times 4 + 0.5 \times 4 = 4$. Thus, the cost $H^{D_3}(G) = 1 \times 4 + 1 \times 4 + (0.5 \times 4 + 0.5 \times 4) + (0.5 \times 4 + 0.5 \times 4) + 1 \times 4 + 1 \times 4 = 24$.

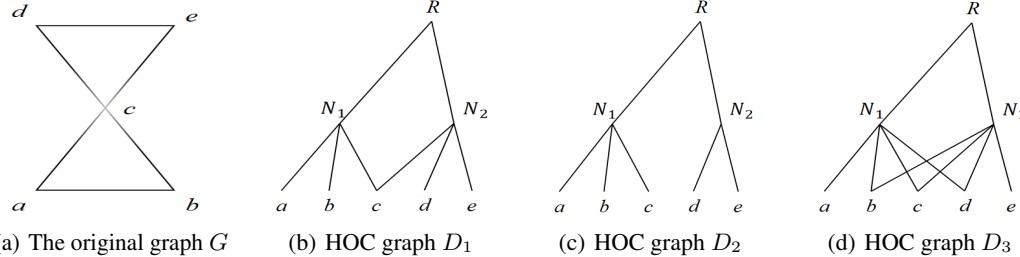

(a) The original graph $G$  (b) HOC graph $D_1$  (c) HOC graph $D_2$  (d) HOC graph $D_3$

*Figure 4.* An example of HOC graphs. The costs $H^{D_1}(G) = 18$, $H^{D_2}(G) = 21$, $H^{D_3}(G) = 24$.

## A.4. Proofs of properties of the cost function

(1) proof of Property 2.11 (compatibility)

*Proof.* When $D$ is an HC tree, the MCA of edge $(u, v)$ is unique and degenerates to the LCA on the HC tree, and the edge-to-node belonging factor is also 1. Then, we get

$$H^D(G) = \sum_{(u,v)\in E} \left( w(u,v) \sum_{N\in M_{uv}} \beta^N_{(u,v)} \cdot |V(N)| \right) = \sum_{(u,v)\in E} w(u,v) \cdot |u \vee v| = \text{Das}^D(G).$$

$\square$

(2) Proof of Property 2.12 (additivity on nodes)

*Proof.* It follows easily by exchanging the summations in the definition of $H^{\mathcal{G}}(\mathcal{D})$.

$$H^D(G) = \sum_{(u,v)\in E} \left( w(u,v) \sum_{N\in M_{uv}} \beta^N_{(u,v)} \cdot |V(N)| \right) = \sum_{N\in D} \left( |V(N)| \cdot \sum_{(u,v)\in E^D_N} \beta^N_{(u,v)} \cdot w(u,v) \right).$$

$\square$

(3) Proof of Property 2.13 (binary optimality)

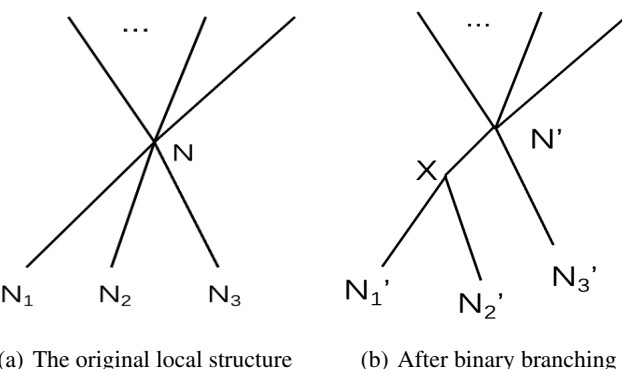

(a) The original local structure  (b) After binary branching

*Figure 5.* Binary optimality of HOC graphs

*Proof.* As shown in Figure 5, assume that (a) represents a local pattern of a non-leaf node on HOC graph $D$, where node $N$ has at least three children $N_1$, $N_2$, and $N_3$. For any edge $(u, v)$ treating $N$ as an MCA, $u$ and $v$ cannot belong

both to any individual cluster of $N_1$, $N_2$ or $N_3$, since otherwise, $N$ would not be the MCA of them. Without loss of generality, let's assume that $u$ belongs to $N_1$ and $v$ belongs to $N_2$ (or $v$ belongs to $N_2$ and $N_3$ simultaneously).

Now, the binary branching operation inserts a new node $X$ as the parent of $N_1$ and $N_2$, which turns into the structure shown in (b). As a result, the MCA for $(u, v)$ becomes $X$. Then we show that after the binary branching, the overall cost $H^D(G)$ does not increase.

After binary branching, since every original path from $u$ to $N$ via $N_1$ passes through $X$ now, and $\alpha_{X,N'} = 1$, we have $\alpha_{u,N} = \alpha_{u,N'} \geq \alpha_{u,X}$. For the same reason, $\alpha_{v,N} \geq \alpha_{v,X}$. Therefore, $\beta^N_{(u,v)} \geq \beta^X_{(u,v)}$. Additionally, we have $|V(X)| < |V(N)|$, implying that $w(u,v)\beta^N_{(u,v)}|V(X)| < w(u,v)\beta^N_{(u,v)}|V(N)|$. This leads to a reduction in the cost function associated with this term. Note that the cost reduction holds for any such edge $(u,v)$ that treats $N$ as an MCA. By Property 2.12, the cost function can be expressed as $\sum_{N \in D} |V(N)| \cdot \sum_{(u,v) \in E^D_N} \beta^N_{(u,v)} \cdot w(u,v)$. The above operation affects only the cost of a single node. For other nodes, the edge-to-node belonging factor and the number of descendant leaf nodes remain unchanged, thus their values do not change. As a result, the overall cost $H^D(G)$ does not increase.. $\qquad \square$

# B. Supplements to the algorithms

In this section, we provide some supplements to our algorithms.

## B.1. Primal and dual problems of $2$-OC

Following is the derivation process of the forms of $cost_{primal}$ and $cost_{dual}$ for 2-OC.

$$
\begin{aligned}
H^D(G) &= \sum_{(u,v) \in E} w(u,v) \sum_{N \in M_{uv}} \beta^N_{(u,v)} \cdot |V(N)| \\
&= \sum_{(u,v) \in E(A)} w(u,v)(|A| + |B|) + \sum_{(u,v) \in E(A,B)} w(u,v)(|A| + |B|) \\
&\quad + \sum_{(u,v) \in E(C)} w(u,v)(|B| + |C|) + \sum_{(u,v) \in E(B,C)} w(u,v)(|B| + |C|) \\
&\quad + \sum_{(u,v) \in E(B)} w(u,v) \left( \beta^{N_1}_{(u,v)} \cdot (|A| + |B|) + \beta^{N_2}_{(u,v)} \cdot (|B| + |C|) \right) \\
&\quad + \sum_{(u,v) \in E(A,C)} w(u,v)n \\
&= w(A)(|A| + |B|) + w(A,B)(|A| + |B|) + w(C)(|B| + |C|) + w(B,C)(|B| + |C|) \\
&\quad + \frac{1}{2}w(B)(|A| + |B| + |B| + |C|) + w(A,C)n \\
&= (w(A) + w(A,B))(|A| + |B|) + (w(B,C) + w(C))(|B| + |C|) \\
&\quad + w(B)\frac{|A| + 2|B| + |C|}{2} + w(A,C)n.
\end{aligned}
$$

Explanation of the derivation: We classify the edges into six parts, denoted by $E(A), E(A,B), E(B,C), E(C), E(B), E(A,C)$, and calculate the cost of each part separately. For $(u,v) \in E(A), E(A,B)$, the only MCA is $N_1$, $\beta^{N_1}_{(u,v)} = 1, |V(N_1)| = |A| + |B|$. Similarly, for $(u,v) \in E(C), E(B,C)$, the only MCA is $N_2$, $\beta^{N_2}_{(u,v)} = 1, |V(N_2)| = |B| + |C|$. For $(u,v) \in E(B)$, the MCAs are $N_1, N_2$, and $\beta^{N_1}_{(u,v)} = \beta^{N_2}_{(u,v)} = \frac{1}{2}$. For $(u,v) \in E(A,C)$, the only MCA is $R$, $\beta^R_{(u,v)} = 1, |V(R)| = n$.

Observe that

$$
|A| + |B|, |B| + |C|, \frac{|A| + 2|B| + |C|}{2} < n,
$$
$$
|A| + |B| + |C| = n,
$$
$$
w(E) = w(A) + w(B) + w(C) + w(A, B) + w(B, C) + w(A, C),
$$
$$
|A| + |B| = n - |C|,
$$
$$
|B| + |C| = n - |A|,
$$
$$
\frac{|A| + 2|B| + |C|}{2} = n - \frac{|A| + |C|}{2}.
$$

Then we have

$$
\begin{aligned}
& cost_{primal}(A, B, C) \\
= \ & (w(A) + w(A, B))(|A| + |B|) + (w(B, C) + w(C))(|B| + |C|) + w(B)\frac{|A| + 2|B| + |C|}{2} + w(A, C)n \\
= \ & (w(A) + w(A, B))(n - |C|) + (w(B, C) + w(C))(n - |A|) + w(B)(n - \frac{|A| + |C|}{2}) + w(A, C)n \\
= \ & nw(E) - \left(w(A) + w(A, B) + \frac{w(B)}{2}\right)|C| - \left(w(C) + w(B, C) + \frac{w(B)}{2}\right)|A| \\
= \ & nw(E) - \left(w(A + B) - \frac{w(B)}{2}\right)|C| - \left(w(B + C) - \frac{w(B)}{2}\right)|A|.
\end{aligned}
$$

### B.2. Proof of Theorem 3.1

We prove Theorem 3.1 with two lemmas respectively.

**Lemma B.1.** *The time complexity of Algorithm 1 is $O(\frac{n^4 \log m}{\epsilon})$.*

*Proof.* Let $w_{max}$ be the maximum edge weight, then $w(E) \leq mw_{max}$, $xw(B) \leq xmw_{max}$, and so $cost_{temp} = w(E) - w(A, C) + xw(B) \leq (1 + x)mw_{max}$. Observe that the edge with the maximum weight is not in $E(A, C)$ in the initial state, and so $cost_{temp} \geq w_{max}$. In each iteration, $cost_{temp}$ increases by $1 + \frac{\epsilon}{n^2}$ times, then the maximum number of iterations is no more than $\log_{1+\frac{\epsilon}{n^2}}((1 + x)mw_{max}/w_{max}) = O(\frac{n^2 \log m}{\epsilon})$. In each iteration, it takes $O(n^2)$ time to calculate the variation caused by the swap of each node pair and update $w(A, C)$ and $w(B)$. Therefore, the overall time complexity is $O(\frac{n^4 \log m}{\epsilon})$. $\qquad\square$

**Lemma B.2.** *The approximation factor of Algorithm 1 is $\frac{2}{3\sqrt{6}} - \Theta(\frac{1+\epsilon}{n})$. That is, A, B, C output by the algorithm satisfy*

$$
cost_{dual} \geq \left(\frac{2}{3\sqrt{6}} - \Theta\left(\frac{1 + \epsilon}{n}\right)\right) cost_{dual}(A^*, B^*, C^*),
$$

*where $A^*$, $B^*$, $C^*$ is an optimal solution.*

*Proof.* Let's consider the status of $A$, $B$, $C$ after the termination of Algorithm 1. Note that exchanging any two nodes from any two of sets $A$, $B$, $C$ at this time cannot make $cost_{temp}(A', B', C')$ larger than $(1 + \frac{\epsilon}{n^2})cost_{temp}(A, B, C)$ any more, where $A'$, $B'$, $C'$ denote the corresponding sets after the exchange, respectively. Let $cost_{old} = cost_{temp}(A, B, C)$. In other words, if an exchange is performed on any pair of nodes, then $cost_{new} \leq (1 + \frac{\epsilon}{n^2})cost_{old}$, where $cost_{new}$ denotes $cost_{temp}$ after the exchange. Define $\Delta = cost_{new} - cost_{old}$, and then we get $\Delta \leq \frac{\epsilon}{n^2}cost_{old}$.

Next, we analyze the $\Delta$ value caused by node exchange. First of all, let's establish some relationships on node swap for later use. For $a \in A$, $b \in B$, $c \in C$, consider the following three cases.

(1) Swap $a$ and $b$: $w(A', C') - w(A, C) = w(b, C) - w(a, C)$, $w(B') - w(B) = w(a, B) - w(a, b) - w(b, B)$.

(2) Swap $b$ and $c$: $w(A', C') - w(A, C) = -w(c, A) + w(b, A)$, $w(B') - w(B) = -w(b, B) + w(c, B) - w(b, c)$.

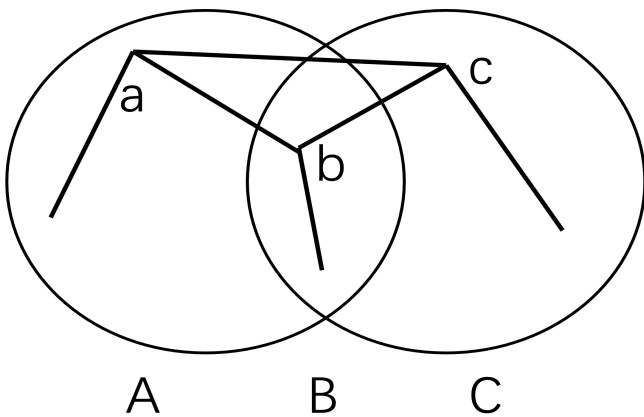

Figure 6. Illustration of 2-OC clustering and node swap

(3) Swap $a$ and $c$: $w(A', C') - w(A, C) = -w(a, C) - w(c, A) + w(a, A) + w(c, C) + 2w(a, c), w(B') - w(B) = 0$.

Recall that $\Delta = cost_{new} - cost_{old} = -(w(A', C') - w(A, C)) + x(w(B') - w(B))$. Therefore, by substituting the above equations, we can obtain the value of $\Delta$ for each case.

(1) For any $a \in A, b \in B$, swapping $a, b$, we have $\Delta = -w(b, C) + w(a, C) + x(w(a, B) - w(a, b) - w(b, B)) \leq \frac{\epsilon}{n^2} cost_{old}$. Summing over all $a, b$, we have

$$-pnw(B, C) + (1 - 2p)nw(A, C) + x\left((1 - 2p)\, nw(A, B) - w(A, B) - 2pnw(B)\right) \leq \frac{\epsilon}{n^2} p(1 - 2p)n^2 cost_{old}. \quad (4)$$

(2) For any $b \in B, c \in C$, swapping $b, c$, by the symmetry of $A, C$, we have

$$-pnw(A, B) + (1 - 2p)nw(A, C) + x((1 - 2p)nw(B, C) - w(B, C) - 2pnw(B)) \leq \frac{\epsilon}{n^2} p(1 - 2p)n^2 cost_{old}. \quad (5)$$

(3) For any $a \in A, c \in C$, swapping $a, c$, $\Delta = w(a, C) + w(c, A) - w(a, A) - w(c, C) - 2w(a, c) \leq \frac{\epsilon}{n^2} cost_{old}$. Summing over all $a, c$, we have

$$pnw(A, C) + pnw(A, C) - 2pnw(A) - 2pnw(C) - 2w(A, C) \leq \frac{\epsilon}{n^2} p^2 n^2 cost_{old},$$

and then

$$-w(A) - w(C) \leq -\frac{pn - 1}{pn} w(A, C) + \frac{\epsilon p^2 cost_{old}}{2pn}. \quad (6)$$

Summing up Inequalities (4) and (5), we get

$$((-p + (1 - 2p)x)n - x) \cdot (w(A, B) + w(B, C)) - 4pxnw(B) \leq (-2 + 4p)nw(A, C) + 2\epsilon p(1 - 2p)cost_{old}. \quad (7)$$

Substituting $x = \frac{p}{1 + 2p}$ into Inequality (7), we get

$$-\frac{4p^2 n + p}{1 + 2p}(w(A, B) + w(B, C) + w(B)) + \frac{p}{1 + 2p} w(B) \leq (-2 + 4p)nw(A, C) + 2\epsilon p(1 - 2p)cost_{old}. \quad (8)$$

Multiplying the coefficient $\frac{4p^2 n + p}{1 + 2p}$ on both sides of Inequality (6), we get

$$-\frac{4p^2 n + p}{1 + 2p}(w(A) + w(C)) \leq -\frac{4p^2 n + p}{1 + 2p} \cdot \frac{pn - 1}{pn} w(A, C) + \frac{4p^2 n + p}{1 + 2p} \cdot \frac{\epsilon p^2 cost_{old}}{2pn}. \quad (9)$$

Summing up Inequalities (8) and (9), we have

$$-\frac{4p^2n+p}{1+2p}w(E) + \frac{p}{1+2p}w(B) \leq \left((-2+4p)n - \frac{4p^2n+p}{1+2p}\cdot\frac{2pn-1}{pn}\right)w(A,C)$$
$$+2\epsilon p(1-2p)cost_{old} + \frac{4p^2n+p}{1+2p}\cdot\frac{\epsilon p^2cost_{old}}{2pn}. \tag{10}$$

After removing $w(B)$ on the left, we get

$$\frac{4p^2n+p}{1+2p}w(E) \geq \left((2-4p)n + \frac{4p^2n+p}{1+2p}\cdot\frac{2pn-1}{pn}\right)w(A,C)$$
$$-2\epsilon p(1-2p)cost_{old} - \frac{4p^2n+p}{1+2p}\cdot\frac{\epsilon p^2cost_{old}}{2pn}. \tag{11}$$

So $w(A,C)$ is upper bounded as

$$
\begin{aligned}
w(A,C) &\leq \frac{\frac{4p^2n+p}{1+2p}w(E) + 2\epsilon p(1-2p)cost_{old} + \frac{4p^2n+p}{1+2p}\cdot\frac{\epsilon p^2cost_{old}}{2pn}}{(2-4p)n + \frac{4p^2n+p}{1+2p}\cdot\frac{2pn-1}{pn}} \\
&\leq \frac{4p^2n^2+pn}{2n^2-2pn-1}w(E) + \frac{2np(-6p^2+p+1)\epsilon}{2n^2-2np-1}w(E) + \frac{p^2(4np+1)(3p+1)\epsilon}{2(2p+1)(2n^2-2np-1)}w(E) \\
&= \left(2p^2+\Theta\left(\frac{1}{n}\right)\right)w(E) + \Theta\left(\frac{\epsilon}{n}\right)w(E) \\
&= \left(2p^2+\Theta\left(\frac{1+\epsilon}{n}\right)\right)w(E). \tag{12}
\end{aligned}
$$

So

$$w(E) - w(A,C) \geq \left(1 - 2p^2 - \Theta\left(\frac{1+\epsilon}{n}\right)\right)w(E).$$

Therefore,

$$
\begin{aligned}
cost_{dual}(A,B,C) &= \left(w(A+B) - \frac{w(B)}{2}\right)|C| + \left(w(B+C) - \frac{w(B)}{2}\right)|A| \\
&= (w(A+B) + w(B+C) - w(B))pn \\
&= (w(E) - w(A,C))pn \\
&\geq \left(1 - 2p^2 - \Theta\left(\frac{1+\epsilon}{n}\right)\right)w(E)pn \\
&= \left(-2p^3 + p - \Theta\left(\frac{1+\epsilon}{n}\right)\right)nw(E) \\
&\geq \left(-2p^3 + p - \Theta\left(\frac{1+\epsilon}{n}\right)\right)cost_{dual}(A^*,B^*,C^*). \tag{13}
\end{aligned}
$$

The fact $cost_{dual} \leq nw(E)$ is used for the last inequality. Substituting $p = \frac{1}{\sqrt{6}}$ into Inequality (13), we get

$$cost_{dual}(A,B,C) \geq \left(\frac{2}{3\sqrt{6}} - \Theta\left(\frac{1+\epsilon}{n}\right)\right)cost_{dual}(A^*,B^*,C^*). \tag{14}$$

$$\square$$

Remarks: We have a rounding error in $cost_{dual}(A,B,C)$ incurred by setting $|A|, |B|, |C|$ to be integers in the above proof. But because the error is additive and aggregated only a constant multiple of $w(E)$, it can be absorbed safely in $\Theta\left(\frac{1+\epsilon}{n}\right)$.

By Lemma B.1, B.2, Theorem 3.1 follows.

### B.3. Proof of Proposition 3.2

*Proof.* We show that the optimal 2-OC-D value $OPT(K_n) = (\frac{2}{3\sqrt{6}} - \Theta(\frac{1}{n}))nw(E)$. Consider any $A$, $B$, $C$, let $x = |A|$, $y = |B|$, $z = |C|$. Let

$$f(x,y,z) = cost_{dual}(A,B,C) = \left(\frac{x(x-1)}{2} + xy + \frac{y(y-1)}{4}\right)z + \left(\frac{z(z-1)}{2} + yz + \frac{y(y-1)}{4}\right)x,$$

and our goal is $\max_{x,y,z\in\mathbb{Z}_+, x+y+z=n} f(x,y,z)$.

First prove that $f(x,y,z)$ takes the maximum value when $x = y$.

Consider $(x,y,z) \to \left(\frac{x+z}{2}, y, \frac{x+z}{2}\right)$, then we have

$$
\begin{aligned}
&f\left(\frac{x+z}{2}, y, \frac{x+z}{2}\right) - f(x,y,z) \\
=\ & \frac{x+z}{2}\left[\frac{x+z}{2}\left(\frac{x+z}{2} - 1\right) + (x+z)y + \frac{y(y-1)}{2}\right] - \frac{x+z}{2}\frac{y(y-1)}{2} - 2xyz - \frac{xz(x+z-2)}{2} \\
=\ & \frac{x+z-2}{2}\left[\frac{(x+z)^2 - 4xz}{4}\right] + \frac{(x+z)^2 - 4xz}{2}y \\
\geq\ & 0,
\end{aligned}
$$

which indicates that when $y$ is given, $f$ gets maximum when $x = z$. So our problem is transformed into

$$\max_{x,y\in\mathbb{Z}_+, 2x+y=n} g(x,y) = \left[(x-1)x + 2xy + \frac{y(y-1)}{2}\right]x,$$

which is equivalent to

$$\max_{0 < x < \frac{n}{2}} h(x) = \left[(x-1)x + 2x(n-2x) + \frac{(n-2x)(n-2x-1)}{2}\right]x.$$

After simplification

$$h(x) = \left[(x-1)x + (n-2x)\frac{n-1+2x}{2}\right]x = \left[(x-1)x + \frac{n^2 - 4x^2 - (n-2x)}{2}\right]x,$$

and taking derivative

$$h'(x) = -3x^2 + \frac{n^2 - n}{2},$$

we know that $h(x)$ achieves the maximum value at $x = \sqrt{\frac{n^2-n}{6}}$. For the optimal solution value, we have

$$OPT(K_n) = -\frac{n^2 - n}{6}\sqrt{\frac{n^2 - n}{6}} + \frac{n^2 - n}{2}\sqrt{\frac{n^2 - n}{6}} = \frac{n^2 - n}{3}\sqrt{\frac{n^2 - n}{6}}.$$

Finally, note that $nw(E) = n \cdot \frac{n(n-1)}{2} = \frac{n^3 - n^2}{2}$. Then we have

$$\frac{OPT(K_n)}{nw(E)} = \frac{\frac{n^2-n}{3}\sqrt{\frac{n^2-n}{6}}}{\frac{n^3-n^2}{2}} = \frac{2}{3\sqrt{6}}\sqrt{\frac{n^2 - n}{n^2}} = \frac{2}{3\sqrt{6}}\left(1 - \frac{1}{n + \sqrt{n^2 - n}}\right) = \frac{2}{3\sqrt{6}} - \Theta\left(\frac{1}{n}\right).$$

This completes the proof of Proposition 3.2. □

## B.4. Proof of Theorem 3.3

In the section, we prove Theorem 3.3 that provides an approximation guarantee for 2-OC-P.

*Proof.* Let $cost^*_{dual}$ and $cost^*_{primal}$ be the optimal objective values, $cost_{dual}$ and $cost_{orimal}$ be the values that Algorithm 1 outputs, for 2-OC-D and 2-OC-P, respectively. Let $a$ be the approximation factor of Algorithm 1 to $nw(E)$ for the dual problem, and $b$ be the approximation factor of Algorithm 1 to $cost^*_{primal}$ for the primal problem. We have the following relationship.

**Lemma B.3.** $b \leq (1-a)(1 + cost^*_{dual}/cost^*_{primal})$.

*Proof.* Note the following relationships hold.

$$cost^*_{primal} + cost^*_{dual} = cost_{primal} + cost_{dual} = n \cdot w(E),$$

and

$$cost_{dual} \geq a \cdot nw(E).$$

Therefore, we have

$$cost_{primal} = nw(E) - cost_{dual} \leq nw(E) - anw(E) = (1-a)(cost^*_{primal} + cost^*_{dual}).$$

That is,

$$cost_{primal}/cost^*_{primal} \leq (1-a)(1 + cost^*_{dual}/cost^*_{primal}).$$

Lemma B.3 follows. $\square$

Since we already have $a = \frac{2}{3\sqrt{6}} - \Theta(\frac{1}{n})$, if we can give an upper bound on $cost^*_{dual}/cost^*_{primal}$, then we also have an upper bound on $b$. The following lemma provides this upper bound.

**Lemma B.4.** *For 2-OC-D and 2-OC-P, $cost^*_{dual}/cost^*_{primal} \leq \rho_{max}/\rho_{avg} \leq d_{max}/d_{avg}$, where $\rho_{avg} = w(E)/|V|$ is the average density, $\rho_{max}$ is the maximum density of all induced subgraphs, $d_{max}$ is the maximum degree of all nodes, and $d_{avg}$ is the average degree of all nodes.*

Note that Lemma B.4 also holds for the dual and primal cost ratio determined by any HOC graph.

*Proof.* Let $A^*$, $B^*$, $C^*$ be the optimal solution (no matter whether it is primal or dual because the two problems are equivalent), let $cost^*_{primal}$ denote $cost_{primal}(A^*, B^*, C^*)$ and $cost^*_{dual}$ denote $cost_{dual}(A^*, B^*, C^*)$ for short.

$$
\begin{aligned}
cost^*_{dual} &= |C^*|\left(w(A^*) + w(A^*, B^*) + \frac{1}{2}w(B^*)\right) + |A^*|\left(w(C^*) + w(B^*, C^*) + \frac{1}{2}w(B^*)\right), \\
cost^*_{primal} &= (|A^*| + |B^*|)\left(w(A^*) + w(A^*, B^*) + \frac{1}{2}w(B^*)\right) \\
&\quad + (|B^*| + |C^*|)\left(w(C^*) + w(B^*, C^*) + \frac{1}{2}w(B^*)\right) + nw(A^*, C^*).
\end{aligned}
$$

Consider two cases of $cost^*_{dual}/cost^*_{primal}$.

(1) If $cost^*_{dual}/cost^*_{primal} \leq 1$, then $b = 2(1-a)$.

(2) If $cost^*_{dual}/cost^*_{primal} \geq 1$, namely $cost^*_{dual} \geq cost^*_{primal}$, substituting into the specific form of the objective function, we have

$$|C^*| \left( w(A^*) + w(A^*, B^*) + \frac{1}{2} w(B^*) \right) + |A^*| \left( w(C^*) + w(B^*, C^*) + \frac{1}{2} w(B^*) \right)$$

$$\geq (|A^*| + |B^*|) \left( w(A^*) + w(A^*, B^*) + \frac{1}{2} w(B^*) \right)$$

$$+ (|B^*| + |C^*|) \left( w(C^*) + w(B^*, C^*) + \frac{1}{2} w(B^*) \right) + nw(A^*, C^*).$$

Then we get

$$(|C^*| - |A^*| - |B^*|) \left( w(A^*) + w(A^*, B^*) + \frac{1}{2} w(B^*) \right)$$

$$+ (|A^*| - |B^*| - |C^*|) \left( w(C^*) + w(B^*, C^*) + \frac{1}{2} w(B^*) \right)$$

$$\geq nw(A^*, C^*).$$

This holds if and only if

$$(|C^*| - |A^*|)(w(A^*) + w(A^*, B^*) - w(C^*) - w(B^*, C^*))$$

$$\geq nw(A^*, C^*) + |B^*|(w(A^*) + w(A^*, B^*) + w(B^*) + w(C^*) + w(B^*, C^*)),$$

which implies that

$$(|C^*| - |A^*|)(w(A^*) + w(A^*, B^*) - w(C^*) - w(B^*, C^*)) \geq 0.$$

Without loss of generality, we assume that $|C^*| \geq |A^*|, w(A^*) + w(A^*, B^*) \geq w(C^*) + w(B^*, C^*)$. We consider two types of scaling for $cost^*_{dual}/cost^*_{primal}$.

Scale 1: replace $|A^*|$ with $|C^*|$ for the numerator, remove $nw(A^*, C^*)$ for the denominator, replace $|C^*|$ with $|A^*|$, and then we get

$$\frac{cost^*_{dual}}{cost^*_{primal}} \leq \frac{|C^*|(w(A^*) + w(A^*, B^*) + \frac{1}{2} w(B^*)) + |C^*|(w(C^*) + w(B^*, C^*) + \frac{1}{2} w(B^*))}{(|A^*| + |B^*|)(w(A^*) + w(A^*, B^*) + \frac{1}{2} w(B^*)) + (|B^*| + |A^*|)(w(C^*) + w(B^*, C^*) + \frac{1}{2} w(B^*))}$$

$$= \frac{|C^*|(w(A^*) + w(A^*, B^*) + w(B^*) + w(C^*) + w(B^*, C^*))}{(|A^*| + |B^*|)(w(A^*) + w(A^*, B^*) + w(B^*) + w(C^*) + w(B^*, C^*))}$$

$$= \frac{|C^*|}{|B^*| + |A^*|}.$$

Scale 2: replace the numerator $w(C^*) + w(B^*, C^*)$ with $w(A^*) + w(A^*, B^*)$, remove the denominator Contents of $|B^*|$, and then we have

$$\frac{cost^*_{dual}}{cost^*_{primal}} \leq \frac{(|A^*| + |C^*|)(w(A^*) + w(A^*, B^*) + \frac{1}{2} w(B^*))}{|A^*|(w(C^*) + w(B^*, C^*) + \frac{1}{2} w(B^*)) + |C^*|(w(C^*) + w(B^*, C^*) + \frac{1}{2} w(B^*)) + (|A^*| + |C^*|)w(A^*, C^*)}$$

$$\leq \frac{(|A^*| + |C^*|)(w(A^*) + w(A^*, B^*) + \frac{1}{2} w(B^*))}{(|A^*| + |C^*|)(w(C^*) + w(B^*, C^*) + \frac{1}{2} w(B^*) + w(A^*, C^*))}$$

$$= \frac{w(A^*) + w(A^*, B^*) + \frac{1}{2} w(B^*)}{w(C^*) + w(B^*, C^*) + \frac{1}{2} w(B^*) + w(A^*, C^*)}.$$

Therefore,

$$\frac{cost^*_{dual}}{cost^*_{primal}} \leq \max \left\{ 1, \min \left\{ \frac{|C^*|}{|A^*| + |B^*|}, \frac{w(A^*) + w(A^*, B^*) + \frac{1}{2} w(B^*)}{w(C^*) + w(B^*, C^*) + \frac{1}{2} w(B^*) + w(A^*, C^*)} \right\} \right\}.$$

Let

$$x = \frac{|C^*|}{|A^*| + |B^*|}$$

$$y = \frac{w(A^*) + w(A^*, B^*) + \frac{1}{2}w(B^*)}{w(C^*) + w(B^*, C^*) + \frac{1}{2}w(B^*) + w(A^*, C^*)}.$$

We have

$$|A^*| + |B^*| = \frac{1}{1+x}|V|,$$

and

$$w(A^*) + w(A^*, B^*) + \frac{1}{2}w(B^*) = \frac{y}{1+y}w(E).$$

Recall that $|C^*| > |A^*|$, $w(A^*) + w(A^*, B^*) > w(C^*) + w(B^*, C^*)$, and observe that the density of the induced subgraph $G[A^* + B^*]$ should be large. Set $\rho_{max} = \max_{U \subseteq V}\left\{\frac{w(E(G[U]))}{|U|}\right\}$ to be the maximum density of the induced subgraph on $G$, $E(G[U])$ to be the edge set of $G[U]$, $\rho_{avg} = \frac{w(E)}{|V|}$ to be the average density of $G$, and then

$$\frac{w(A^*) + w(A^*, B^*) + \frac{1}{2}w(B^*)}{|A^*| + |B^*|} = \frac{y(1+x)}{1+y} \cdot \frac{w(E)}{|V|} \le \frac{w(A^* + B^*)}{|A^*| + |B^*|} \le \rho_{max}.$$

We have

$$\frac{y(1+x)}{1+y} \le \frac{\rho_{max}}{\rho_{avg}}.$$

Now according to the value of $min(x, y)$, we consider the following two cases.

(1) $x \le y$, we have

$$x = \frac{(1+y)x}{1+y} = \frac{x+xy}{1+y} \le \frac{y+xy}{1+y} = \frac{y(1+x)}{1+y} \le \frac{\rho_{max}}{\rho_{avg}}.$$

(2) $x > y$, we have

$$y = \frac{y(1+x)}{1+x} < \frac{y(1+x)}{1+y} \le \frac{\rho_{max}}{\rho_{avg}}.$$

Therefore,

$$\min\left\{\frac{|C^*|}{|A^*| + |B^*|}, \frac{w(A^*) + w(A^*, B^*) + \frac{1}{2}w(B^*)}{w(C^*) + w(B^*, C^*) + \frac{1}{2}w(B^*) + w(A^*, C^*)}\right\} \le \frac{\rho_{max}}{\rho_{avg}}.$$

Then for $cost^*_{dual}/cost^*_{primal}$, we have

$$\frac{cost^*_{dual}}{cost^*_{primal}} \le \max\left\{1, \frac{\rho_{max}}{\rho_{avg}}\right\} = \frac{\rho_{max}}{\rho_{avg}}.$$

Calculating $\rho_{max}$ is difficult. However, it can be observed that the average degree $d \le d_{max}$ on $G[U]$, and $w(E(G[U])) = \frac{d|U|}{2}$. So, we have an upper bound on $\rho_{max}$, that is

$$\rho_{max} = \frac{w(E(G[U]))}{|U|} \le \frac{d \cdot |U|}{2|U|} \le \frac{d_{max}}{2}.$$

On the other hand,

$$\rho_{avg} = \frac{w(E)}{|V|} = \frac{d_{avg} \cdot |V|}{2|V|} = \frac{d_{avg}}{2}.$$

This implies that

$$\frac{\rho_{max}}{\rho_{avg}} \leq \frac{d_{max}}{d_{avg}}.$$

Therefore, $\frac{cost^*_{dual}}{cost^*_{primal}} \leq \frac{\rho_{max}}{\rho_{avg}} \leq \frac{d_{max}}{d_{avg}}$, and this completes the proof of Lemma B.4. □

Combining Lemmas B.3 and B.4, Theorem 3.3 follows.

□

## C. Supplements to Experiments

In this section, we provide supplementary information and results on our experiments.

### C.1. Definition of OSBM

OSBM is specified by a pair of real numbers $p_1, p_2$ ($0 \leq p_1 \leq p_2 \leq 1$), and a $k \times k$ symmetric membership matrix $Z$, in which each row or column indicates a cluster and each entry is a natural number. $Z_{ij}$ ($i \neq j$) represents the number of overlapping nodes between the $i$-th and the $j$-th clusters, and $Z_{ii}$ represents the number of nodes in the $i$-th cluster that do not present overlap. Denote by $C_1, ...C_k$ the planted overlapping clusters. $p_1$ represents the inter-link probability between each pair of clusters, while $p_2$ represents the intra-link probability within each cluster. For two nodes in the overlapping region, we have two independent samples, and the edge exits if either of the samples generates one. Equivalently, the probability of edge presence between any two nodes in the overlapping region is $1 - (1 - p_2)^2$.

### C.2. Definition of the NMI for OC

The normalized mutual information (NMI) was originally developed as a distance measure for non-overlapping partitions. The work in (McDaid et al., 2011) put forward the NMI for OC that represents a natural generalization from the original NMI. Formally, for two different groups of overlapping clusters $X = \{x_1, x_2, ...\}, Y = \{y_1, y_2, ...\}$ on the same graph $G = (V, E)$, where $x_i, y_i$ are clusters, let $p(x_i) = |x_i|/|V|$, $p(y_i) = |y_i|/|V|$, $p(x_i, y_j) = |x_i \cap y_j|/|V|$. It should be noted that, unlike the classic NMI used for non-overlapping partitions, $p(x_i)$ and $p(y_i)$ cannot be interpreted as probability distributions due to the overlaps within $X$ and $Y$. We define the entropy of random variables, joint entropy and mutual information respectively as follows:

$$H(X) = -\sum_{x_i \in X} p(x_i) \log p(x_i), \ H(Y) = -\sum_{y_i \in Y} p(y_i) \log p(y_i),$$

$$H(X, Y) = -\sum_{x_i \in X, y_j \in Y} p(x_i, y_j) \log p(x_i, y_j),$$

$$I(X : Y) = H(X) + H(Y) - H(X, Y).$$

Then the NMI of $X, Y$ is defined as

$$NMI(X, Y) = \frac{I(X : Y)}{\max(H(X), H(Y))}.$$

$NMI(X, Y)$ is in the range $[0, 1]$, and equals 1 if and only if $X$ and $Y$ are exactly coincident.

### C.3. Evaluation on the MNIST dataset

To show intuitively that our algorithm is able to find out the blurred overlapping area of datasets, we run our 2-OC algorithm on the MNIST dataset (LeCun et al., 1998), which is a benchmark of handwritten digits containing ten classes of images labeled by $0 \sim 9$, respectively. We select two pairs of labels that are easily confused by hand writing, i.e., 1 vs. 7, 3 vs. 8, and construct a $k$-nearest neighbor graph for each of them. Each node of the graph represents an image of handwritten digit, and the similarity is measured by applying the Gaussian kernel function to the Euclidean distance of pixel vectors. We remark that not all embeddings (e.g., word embeddings) that are generated by modern-day AI models are suitable for

clustering. We just find that pixel vector in MNIST is somewhat a good use-case to showcase our results of overlapping, ambiguous samples.

The parameters, NMI, size of the overlapping part, the costs of ground truth (GT, non-overlap) and 2-OC output are summarized in Table 5. NMI is calculated with the non-overlapping ground truth of data points, although our algorithm gives overlapping results. However, the NMI for the labels 1 vs. 7 is above 0.9, and only 4 digits, which can be viewed as ambiguous ones, are allocated in the overlapping part. We demonstrate all of them in Figure 7(a). For the labels 3 vs. 8, there are 60 ambiguities. We demonstrate four of them in Figure 7(b). A significant factor that impacts the accuracy of our algorithm is that we simply use the pixel vectors of digits which is a very rough representation of images.

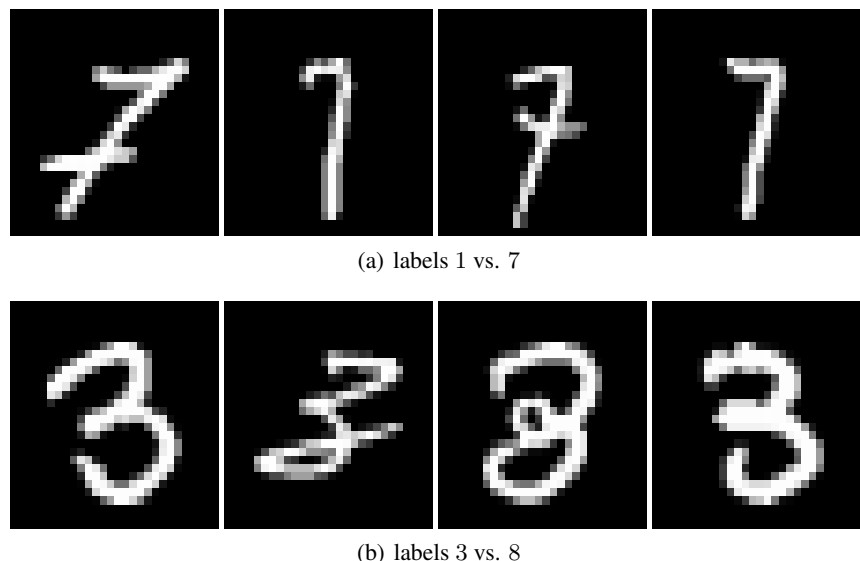

(a) labels 1 vs. 7

(b) labels 3 vs. 8

*Figure 7.* Demonstrations of the ambiguous samples our 2-OC algorithm yields.

Table 5. Parameters and results on the MNIST dataset

| label | size | $k$ | NMI | overlapping size | GT cost (non-overlap) | 2-OC cost |
|-------|------|-----|-----|------------------|----------------------|-----------|
| $1\,vs.\,7$ | $7877 + 7293$ | 100 | 0.926 | 4 | $7.37 \times 10^9$ | $7.33 \times 10^9$ |
| $3\,vs.\,8$ | $7141 + 6825$ | 100 | 0.812 | 60 | $6.47 \times 10^9$ | $6.54 \times 10^9$ |

## C.4. Visualization of four overlapping clusters

We visualize in Figure 8 a 4-HOC results of Algorithm 2 on a small graph that is generated from OSBM and contains 100 nodes and 629 edges. It has four embedded overlapping clusters of size 27, each of which contains 22 or 24 nodes that entirely belong to the cluster. There are 6 overlapping regions, each of which corresponds to a pair of overlapping clusters out of the 4 ones, and each region contains 2 nodes. We label them from 91 to 100. We demonstrate the ground-truth membership of all nodes in Tables 6 and 7. The edge presence probabilities are $p_1 = 0.01$, $p_2 = 0.05$ and $p_3 = 0.3$.

*Table 6.* Membership of level-2 nodes in each of the four clusters. Each diagonal entry numbers the nodes that belong exclusively to the corresponding cluster. The entry $(i, j)$ $(i \neq j)$ denotes the node numbers in the overlapping region between clusters $i$ and $j$. In the visualization, the corresponding colors of clusters 1, 2, 3, and 4 are red, green, blue, and yellow , respectively, while the overlapping nodes are the mixed colors of their clusters.

| cluster label | 1 | 2 | 3 | 4 |
|---------------|------|------|-------|--------|
| 1 | 1-22 | 96 | 91,97 | 92,98 |
| 2 | 96 | 23-44 | 93,99 | 94,100 |
| 3 | 91,97 | 93,99 | 45-66 | 95 |
| 4 | 92,98 | 94,100 | 95 | 67-90 |

*Table 7.* Membership of level-1 nodes in each of the two clusters. Clusters 1 and 2 form one cluster, denoted by $(1, 2)$, on level 1, while clusters $3$ and $4$ form the other one, denoted by $(3, 4)$.

| cluster label | (1,2) | (3,4) |
|---|---|---|
| (1,2) | 1-44,96, | 91-94, 97-100 |
| (3,4) | 91-94, 97-100 | 45-90,95 |

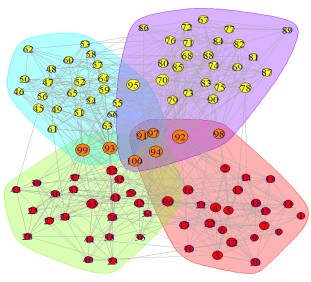

(a) Ground truth of the $4$ clusters (Level 1).

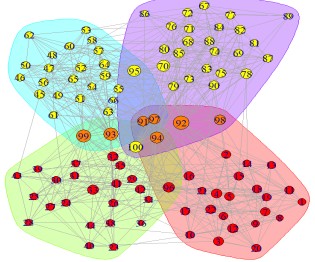

(b) The result of our algorithm (Level 1).

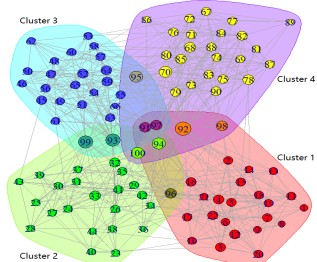

(c) Ground truth of the $4$ clusters (Level 2).

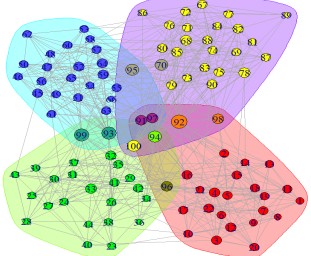

(d) The result of our algorithm (Level 2).

*Figure 8.* Visualization of a $4$-HOC clustering.

*Table 8.* List of misclassified nodes given by our algorithm on level 1.

| node number | cluster label in ground truth | cluster label by the $k$-HOC algorithm |
|---|---|---|
| 100 | (1,2),(3,4) | (3,4) |

*Table 9.* List of misclassified nodes given by our algorithm on level 2.

| node number | cluster label in ground truth | cluster label by the $k$-HOC algorithm |
|---|---|---|
| 100 | 2,4 | 4 |
| 70 | 4 | 3,4 |

Our algorithm bipartitions the node set at the first level into two overlapping clusters, one consists of red and green (1 and 2), the other blue and yellow (3 and 4). It achieves NMI $= 0.964$ on this level. At the second level, it achieves NMI $= 0.959$ for the four ground-truth overlapping clusters. In Figure 8, we visualize this result. Our algorithm successfully captures the overall outlines of the clusters, except membership errors on only two nodes, whose labels are 70 and 100. We list them in Tables 8 and 9. The node 100 is misclassified into non-overlapping region on the first level while the node 70 is misclassified into overlapping communities on the second level.

