# OpenReview forum: "Hierarchical Overlapping Clustering on Graphs: Cost Function, Algorithm and Scalability"
_ICML.cc/2025/Conference — ICML 2025 poster_

### Official Review · Reviewer_7SG8 · 2025-03-04

**Overall Recommendation:** 3

**Summary:**

This paper studies hierarchical overlapping clustering (HOC), in which vertices are assigned to a hierarchical structure of overlapping clusters. In comparison with non-overlapping HC, we construct a DAG rather than an HC tree.

The paper introduces an objective function for this problem, generalising Dasgupta's cost function for HC, and gives a constant-factor approximation algorithm for the dual objective. Finally, the paper includes some experimental evaluation and compares the algorithm with non-hierarchical overlapping clustering algorithms, finding that the new algorithm has a faster running time.

**Claims And Evidence:**

The paper claims three contributions:

1. The introduction of a new cost function for hierarchical overlapping clustering. This is a new problem, not previously studied and so the introduction of such a cost function is quite original. The given objective function extends Dasgupta's cost in a sensible way to the case with overlapping clusters.
2. An approximation algorithm for the dual of the proposed cost function. This is theoretically justified and the claim stands as stated. However, I feel that it is not particularly significant given that the approximation is based on universal bound on the cost of any output produced by the proposed algorithm. That is, there is no theoretical proof that the algorithm performs better on instances with a better optimal cost.
3. The scalability of the proposed algorithm is demonstrated through empirical evaluation. The evaluation justifies the claim, although the tested datasets are of relatively small size (up to 10000) and so it is not clear that the proposed algorithm applies in practical scenarios.

**Essential References Not Discussed:**

No.

**Experimental Designs Or Analyses:**

See the answers above.

**Methods And Evaluation Criteria:**

My only concern with the evaluation criteria is the size of the tested datasets. It would be interesting to see if the proposed algorithm could scale to 100,000 vertices, or more.

**Other Comments Or Suggestions:**

The text in figure 2 is unreadable - it would be better to make it bigger.

**Other Strengths And Weaknesses:**

I feel that the proposed cost function is interesting, but the key weakness of the paper is in the theoretical results. It would be interesting to prove a guarantee on the performance of the algorithm that depends on the optimal cost function of a specific graph.

Edit:
Based on the responses of the authors, I have raised my score.

**Questions For Authors:**

N/A

**Relation To Broader Scientific Literature:**

The results build on the cost function for hierarchical clustering proposed by Dasgupta, and generalise this in a reasonable way to the problem of detecting overlapping clusters, which has been studies only in the non-hierarchical setting.

**Theoretical Claims:**

The theoretical claims appear sound, although I did not check the proofs in detail.

---

> ### Author Rebuttal · Authors · 2025-04-01
>
> We thank the reviewer for the careful review and valuable comments. Let us address the concerns one by one.
>
> 1. On the scalability of our algorithm, we have demonstrated it on graphs of size 100,000. Please note that the largest graph in the second row of Figure 2 has size $10\times 10^4=10^5$ (sorry for the tiny text, we will amplify it in our updated version). We did not scale to larger graphs (e.g., 500,000, or 1,000,000) not because our algorithm cannot bear the large scales, but only because the baseline methods cannot, which results in lacking comparison objects for us. That is also why, as shown in Table 1, in our PC environment, we cannot compare with any baseline on large real-world datasets, but just plant in the last column the results from (Orecchia et al., 2022) whose operating environment includes a cluster of machines.
>
> 2. Regarding the weak theoretical results, we think that as the first step on the hierarchical overlapping clustering (HOC) study, a constant approximation factor (although not very large) for the dual $k$-HOC problem is not so bad. As a by-product of our theory, our study on 2-OC (which is ever one of the central topics on overlapping clustering before our work) achieves theoretical guarantees on both primal and dual problems. The main obstacle of getting a better guarantee for $k$-HOC stems from the complicated hybrid structure during the recursions of the 2-OC algorithm. Further investigating this process could be an interesting next step on $k$-HOC study. We believe that there is better guarantee since the actual performance of our $k$-HOC as shown in Figure 2 is almost perfect on synthetic datasets.
>
> We thank the reviewer again and are happy to address any remaining concerns.

---

> > ### Comment · Reviewer_7SG8 · 2025-04-04
> >
> > Thanks for your responses - I realise that I missed some of the larger experiments and I will increase my score accordingly.

---

> > > ### Author Response · Authors · 2025-04-04
> > >
> > > Thank you for your response and for helping to raise the score. We truly appreciate your support.

---

### Official Review · Reviewer_4xK8 · 2025-03-10

**Overall Recommendation:** 4

**Summary:**

The paper formally introduces the problem of hierarchical overlapping clustering. Overlapping and hierarchical clusterings have been studied more extensively separately. The only preexisting works that have studied them together have been in the distance setting (edge weights are distances) with no formal objectives and guarantees. This paper provides those formalities in the similarity setting (edge weights are similarities).

In hierarchical clustering, clusters are nested into clusters, creating a tiered structure of clusters which can be represented as a tree (the root = the set of all data, the leaves = individual data, and intermediate nodes = clusters). With overlaps, any cluster can have partial belonging to another cluster up the hierarchy. The total belonging (sum over belongings of a single cluster to all other clusters) is, naturally, 1. It has a nice probabilistic interpretation: If I 1/2 belong to one cluster, then maybe I flip a coin to determine my belonging. The hierarchies of belongings are designed such that these probabilities are preserved (if I 1/2 belong to X and 1/2 to Y, and X 1/2 belongs to Z but Y doesn't belong to Z, then I 1/4 belong to Z).

Their objective function directly extends Dasgupta's cost objective, perhaps the most famous and natural hierarchical clustering objective. Effectively, this objective deems that highly similar points should be tightly packed into small clusters. In the overlapping setting, partial belongings are accounted for. A minimum common ancestor of two points is defined as a cluster containing both points whose children contain at most one of the points each. In Dasgupta's setting, this would be the size of the smallest cluster containing the points, but there may be multiple "MCAs" with overlaps. With Dasgupta, the cost contribution of $(x, y)$ is $w(x, y)$ scaled by the size of the MCA, but in this paper, it's a convex combination of the sizes of the many MCAs (again, with a nice probabilistic interpretation). They also define the dual to this objective, which is akin to the work of Moseley and Wang in the hierarchical (non-overlapping) setting.


The authors then propose two algorithms. The first is a simple local search algorithm for the problem of (non-hierarchical) overlapping-clustering with only 2 overlapping clusters allowed. While this problem has been studied before, it is new in the context of how their objective simplifies to this setting. The algorithm achieves an O(1)-approximation for the dual and an O(d_ratio)-approximation for the original problem, where d_ratio is the maximum degree divided by the average degree.

The second algorithm solves the general problem when the width of the hierarchy is bounded by k (except at the leaves). This shown only to approximate the dual, but it does so with the same factor as the previous algorithm and on the more general case.

Finally, they run the latter algorithm against a set of baseline algorithms. This includes one of the previous papers on hierarchical overlapping clustering. The algorithms are run on data generated by an overlapping stochastic block model and on real snap datasets. The latter does not have ground truths for hierarchical clustering, so they only test runtime. On synthetic data, they show that output much better clusterings (as measured by NMI, which they don't define). On all datasets, they show their algorithm is acceptably scalable.

**Claims And Evidence:**

Yes, they are substantiated by accompanied proofs in the appendix as well as formal experiments. I would say the experiments are somewhat limited in scope, partly due to the difficulty of finding real datasets for hierarchical clusterings with ground truth. However: why not compute your proposed dual objective scores for all tested datasets and see how the algorithms compare?

**Essential References Not Discussed:**

None that are essential.

**Experimental Designs Or Analyses:**

The experimental designed seemed okay to me, though, as mentioned, there may be more tests they could run.

**Methods And Evaluation Criteria:**

Yes, though more testing could have been done (see my question about more options).

**Other Comments Or Suggestions:**

I basically formulated these in the "Questions" section. There are typos/weird phrasings around, but I did not write them down. It doesn't significantly impede comprehension.

**Other Strengths And Weaknesses:**

Strengths
1. Very interesting niche that should be studied
2. Nice extensions of seminal works by Dasgupta and Moseley & Wang
3. Shows some nice simple algorithms for this problem
4. Solid set of experiments

Weaknesses
1. I have significant qualms with the proposed model, all of which can be found in the "Questions" section
2. Algorithm 1 is extremely limited (flat overlapping clustering, only 2 clusters). It doesn't even consider the hierarchical aspect, which seems central to this paper. That being said, it is interesting in useful, but it's the only algorithm they propose that approximates their proposed objective as opposed to it's dual, and it's rather disappointing in its limitation.
3. Experiments could be extended

Ultimately, I believe this paper should be published since it initiates an interesting field of study. It also seems to do it in the "right way" with its foundations in the Dasgupta paper, however, the formulation and results are weak. I lean towards acceptance, but am not sure if this truly meets the bar for ICML.

**Questions For Authors:**

1. Are (3) and (4) necessary for Property 2.6? I would think they would be implied by the fact that S is an anti-chain and N in S is ordered between X and Y.
- EDIT:I see (1) and (2) don't cover the "maximality" aspect, but can't you just say it's a maximal set that satisfies those two properties? I think that would be much more natural

2. Along the same lines, shouldn't the statement be "For two nodes X and Y... for any node set S..." Instead of "if" there is a node set S? There would only not exist such a set if they were parent/child right?

3. Consider the example from your paper Fig 1(b). Here, c is assigned 1/2 to both N1 and N2. Am I correct to say that there is only one MCA of (b, c), which is N1, and only one MCA of (c, d), which is N2? Then, the contribution of cost from these edges are just $2w(b, c)$ and $2w(c, d)$ respectively, which is the minimum contribution possible. This feels like cheating to me - shouldn't the partial assignment of c to N2 hurt how much (b, c) contributes to the cost?

For instance, say we removed the edges (a, c) and (b, c) from the graph 1(a) entirely. Shouldn't there be some negative impact on the cost, now that c is partially assigned to N1 and N2? It SHOULD just be assigned to N1, but it doesn't seem like a loss to add another assignment in this case. This doesn't seem like a desirable property.

I think this issue is something you're hiding behind the example 1(d). You're trying to say there IS  a cost to having too many assignments, but that only happens when pairs of vertices are given multiple overlapping assignments unnecessarily, since it increases the number of MCAs there are for an edge. Ultimately, I think my problem here is how you've defined MCA: it feels like N1 AND R should both be partially MCAs of (b, c) in 1(b), since c's assigned partially to N1 and N2.

Note: The issues here are not resolved by bounding the width of the graph.

4. You claim that the longest anti-chain blocks all paths from leaves to root. Is this actually true? Consider the HC tree which is a leafy stick (a and b merge, then they merge with c, then d, ...). The longest anti-chain here is just a single node! Unless, that is, if you include leaves in the anti-chains (but they aren't included in the width, so it seems weird...) You might want to require that there is some anti-chain that DOES do this (e.g., the children of the nodes in the chain consist of precisely the set of leaves).
- EDIT: Is this actually covered by condition (3) in definition 2.1? I may have just missed this

5. Dual HOC: Can you briefly mention the Moseley and Wang dual when you define this on page 5? Since this is the corresponding problem in the HOC setting.

6. In Proposition 3.2, you talk about using approximation methods based off the max potential dual, n*w(e). In HC, Moseley and Wang did a lot of analysis with this max potential dual, and there are similar limitations (I think it's 1/3 or 1/2). It would be great to find where this happens and cite it here! I can't remember off the top of my head .

7. Can you give some intuition as to why the algorithm approximation for the primal is dependent on the degree ratios?

8. I don't know much about stochastic block models - are they hierarchical in structure? I would think you'd want a hierarchical one to show what happens when you're not just representing a flat clustering.

9. What is NMI?

10. Why not compute your proposed dual objective scores for all tested datasets and see how the algorithms compare?

**Relation To Broader Scientific Literature:**

This is the first proposal of a formal objective for hierarchical overlapping clustering. It is notably defined off a seminal work by Dasgupta on regular hierarchical clustering. This is certainly an interesting area that I am surprised has not been studied this rigorously before. It could certainly open up an entirely new niche of study, though I do have qualms with their formulation (discussed later).

**Theoretical Claims:**

I did not verify the correctness of proofs in the appendix, but some of them were plain to see how they worked and the others are certainly plausible. Proofs were mentioned to be in the appendix wherever applicable.

---

> ### Author Rebuttal · Authors · 2025-04-01
>
> We thank the reviewer for the careful review and valuable comments. Let us address the concerns one by one.
>
> Q1: Yes, these four conditions of Property 2.6 can be simplified since (3) implies that every $N\in S$ is ordered between $X$ and $Y$. However, we cannot say that it's a maximal set that satisfies (1) and (2) only since (4) is necessary (this also relates to Q4, and sorry for inaccurate statement therein). Please refer to the toy example in Figure 3, Appendix A.2. Nodes $N_1$ and $N_5$ form a maximal anti-chain between $b$ and $R$ satisfying (1) and (2), and the belonging factors of $b$ to them are $1/2$ and $1/4$, respectively. But $b$, $N_1$ and $N_5$ all belong totally to $R$, which makes Property 2.6 abort (rhs=3/4). The reason is that the path $(b, N_2, N_4, R)$ leaks even though $\{N_1, N_5\}$ is a maximal anti-chain, while (4) blocks this leak.
>
> Q2: Yes, the condition after “if” is quite natural, “for any node set S” is better.
>
> Q3: Note that in Fig 1(b), the contribution of cost from $(b,c)$ and $(c,d)$ are $3w(b,c)$ and $3w(c,d)$, respectively, since $|N_1|=|N_2|=3$. By our definition, indeed, the partial assignment of $c$ to $N_2$ doesn’t hurt the cost that $(b, c)$ contributes to. This is because $(b, c)$ belongs entirely only to $N_1$. When $(a, c)$ and $(b, c)$ are removed from $G$, the cost of any HOC will certainly decrease since the edge number becomes less. But now the optimal HOC should be of shape $D_2$ (exchanging labels of $a, b$ and $d, e$) since the MCA of $(a,b)$ has size only $2$. In this case, $D_1$ indeed has negative impact on the cost when compared to $D_2$. This is quite natural and desirable, isn’t it? (Maybe you just have some misunderstanding in calculating the cost of $D_1$).
>
> As for $D_3$, we indeed want to say there is a cost to having too many assignments. But this cost mainly comes from enlarging the clusters themselves after excessive assignments. As long as you agree with our setting that a cluster can be treated as an MCA of an edge only if the edge is entirely contained in the cluster, then everything will be easy to understand. This setting is consistent with Dasgupta’s cost for HC. For example, In Fig 1(c), we do not treat $N_2$ as the MCA of $(c, d)$ even though $d$ belongs to $N_2$. Note that although the assignment of an endpoint of an edge does not impact on a cluster that is not an MCA, it does impact on the belonging factors to all MCAs. As for your doubt that $N_1$ and $R$ should both be partially MCAs of $(b, c)$ in Fig. 1(b), now that we have defined MINIMAL common ancestor, the partial order between two CAs $N_1$ and $R$ should be avoided.
>
> Q4: The statement here is indeed not accurate. Our original intention is to point out the analogue of significance between the longest anti-chains on HC and HOC. But as shown by the instance in our response to Q1, not any maximal anti-chain necessarily blocks all paths from leaves to the root. We believe that a slight modification of this instance will give evidence to refute a longest anti-chain also. The leaf stick that you propose is another counter example and does not violate condition (3) in Def. 2.1. We can only say that a longest anti-chain is located, intuitively, as far down from the root as possible to block leaf-to-root paths. Thanks for pointing out this.
>
> Q5: Sure. We will mention Moseley and Wang’s dual for HC when we define $k$-HOC dual on Page 5.
>
> Q6: Yes, Moseley and Wang proved a similar limitation of $1/3$ off the trivial upper bound $n*w(e)$ for the binary HC clustering. We will briefly mention it in the discussion of Th. 3.1 in our updated version.
>
> (Since there is a strict limit 5000 on the character number, we continue our response in the rebuttal to Reviewer 3QxV on the top.)

---

> > ### Comment · Reviewer_4xK8 · 2025-04-02
> >
> > Thank you for your responses. I am particularly happy with your response to Q3 (though I still think the fact that R isn't considered an MCA isn't theoretically "nice"). The theoretical results of the algorithm are still weak, but I think this paper is much more holistic than the results of the algorithm. Therefore, I think that the paper deserves acceptance.

---

> > > ### Author Response · Authors · 2025-04-03
> > >
> > > Thank you very much for your recognition for our work. It is quite normal that different people have different opinions. We are happy to see multi-views of hierarchical overlapping clustering, which also sparks our new thinking to this problem. Thank you for your careful reading and valuable comments again.

---

### Official Review · Reviewer_sxS7 · 2025-03-13

**Overall Recommendation:** 3

**Summary:**

This work introduces and studies the hierarchical overlapping clustering (HOC) problem. In the clustering literature, many works have focussed on either (i) overlapping or (ii) hierarchical clustering; this work's aim is to reconcile both topics.

As a first contribution -- inspired by the well-known Dasgupta cost function -- this paper introduces a cost function for overlapping hierarchical clustering. The proposed cost function has serveral desirable properties such as compatibility, additivity, and binary optimality.

As a second contribution, the paper further proposes approximation algorithms for the dual and primal variants of the new cost function. These approximation algorithms hold for a restricted variant of the HOC problem, namely the $k$-HOC problem and the $2$-HOC problem.

As a final contribution, the algorithm is tested on synthetic and real-world dataset. To speed up the algorithm, the paper uses local search heuristics.

**Claims And Evidence:**

Yes

**Essential References Not Discussed:**

N/A

**Experimental Designs Or Analyses:**

The experiment section is slightly discoupled from the main theoretical body of the work

**1** The proposed algorithms are rather slow ($O(n^4)$ factor in the running time) which prohibits the algorithm from being applied at a large scale. This requries the paper to introduce local search heuristics to make the algorithms implementable. As such, Algorithm 2 is not 'formally' compared.

**2** $k$-HOC experiments are only performed on synthetic data, and it is not clear to me how some metrics (such as NMI) are computed for overlapping settings for both methods compared.

**3** No qualitatative results on real-world datasets; the real-world datasets are only used for scalability experiments. There are experiments on MNIST in the appendix, however the NMI in those experiments is calculated using non-overlapping clusters.

**Methods And Evaluation Criteria:**

Yes

**Other Comments Or Suggestions:**

N/A

**Other Strengths And Weaknesses:**

**S1)** The paper studies a new important problem, that is worth to study. The problem formulation itself is quite intuitive, and a natural way to combine the hierarchical and overlapping clustering problems. This work could potentially lay the groundwork for future algorithms to be developed on. Throughout the main body and appendix the paper also states and proves intuitive properties of the cost function, which is a nice contribution.

**S2)** The paper also introduces 2 approximation algorithms for the new objective function, one for the $2$-OC problem, and another $k$-HOC problem. These algorithms could provide a good starting point for future work on HOC.

**S3)** Finally, the theory for the cost function and the proofs for the approximation guarantees seem sound to me - and non-trivial for the most part.


**W1)** The approximation guarantees are given on fairly restricted settings; for the primal and dual variants of $2$-OC and $k$-HOC. the $2$-OC problem studies the overlapping bipartition problem, and contains no hierarchical structure. The $k$-HOC problem does have a hierarchical component, but restricts itself to at most $k$ clusters/nodes. The approximation guarantee on the latter is only given on the dual variant of the problem (which in general is a bit easier to prove, like the dual of Dasgupta's cost function (Mosely & Wang, 2017)). It seems that the approximation guarantee follows rather directly from the approximation guarantee of the dual of $2$-OC.

**Questions For Authors:**

N/A

**Relation To Broader Scientific Literature:**

The paper effectively positions its contributions within existing literature on hierarchical and overlapping clustering, referencing Dasgupta (2016), Orecchia et al. (2022), and other foundational works.

**Theoretical Claims:**

Yes, I checked them.

---

> ### Author Rebuttal · Authors · 2025-04-01
>
> We thank the reviewer for the careful review and valuable comments. Let us address the concerns one by one.
>
> 1. Regarding the speed-up version that makes the comparison of Algorithm 2 not “formal”, let’s look at the two strategies. The first is a good initialization with two non-overlapping clusters and using “move” instead of “exchange” during local search. We don’t think this harms the results of Algorithm 2 much for the following reason. Let’s consider another process as follows. We proceed Algorithm 2 rigorously first until the exchange-based local search gets stuck. By Theorem 3.6, we get our approximation guarantee $\frac{2}{3\sqrt{6}}-\Theta(\frac{1+\epsilon}{n})$. Then we go on to move nodes following the "move" strategy. Since we move nodes only if the cost gets better, when all nodes get stuck, we have no worse cost than that we have from Algorithm 2, or say, this is also an approximation algorithm with the same factor $\frac{2}{3\sqrt{6}}-\Theta(\frac{1+\epsilon}{n})$. This looks like we have started our local search from a good overlapping initial clustering. The only difference of our strategy is that, in practice, we start our local search from a good non-overlapping clustering, which doesn’t seem like to be as good. However, the almost perfect NMI results in Figure 2 demonstrate that even this strategy that should not seem so good is not bad at all.
>
> The second strategy is batch migration. In Algorithm 2, the nodes are exchanged one by one until get stuck. Now we move more than one node in each iteration. This strategy makes no difference in effectiveness from the original algorithm, because the final state of both process is “getting stuck”. Since the approximation guarantee holds for any initialization, any state that makes all nodes stuck can be viewed as a terminating state with this guarantee. So even Algorithm 2 itself can use batch migration without any loss in approximation guarantee.
>
> 2. The definition of NMI metric for overlapping clustering has been provided in Appendix C.3. We have also prompted its place in the paragraph of “Datasets and evaluation” (Page 7) in the main text.
>
> 3. Regarding no qualitative results on real-world datasets, we evaluate the effectiveness of our algorithm on synthetic datasets. Due to lack of ground truth on real-world datasets, we are not able to verify the quality of our clustering on them. Nevertheless, we can compute the objective scores as the Reviewer 4xK8 suggested (Q10 therein). However, there is still no way for us to compare it with the baseline methods since no baseline can terminate within a reasonable time on large graphs when run on a personal computer. This also demonstrates the priority of our algorithm in scalability.
>
> 4. Regarding W1, we think that 2-OC is the foundation of $k$-HOC, just like that bipartition is the foundation of approximate algorithms of hierarchical clustering. We have paid much attention to 2-OC for both dual and primal versions, and achieved approximation guarantees. Yes, we have not achieved an approximation guarantee for the primal $k$-HOC problem since we have not found a proper way to analyze recursions of 2-OC. We think primal $k$-HOC needs more novel insights and is an interesting open problem.
>
> We thank the reviewer again and are happy to address any remaining concerns.

---

### Official Review · Reviewer_3QxV · 2025-03-15

**Overall Recommendation:** 4

**Summary:**

Two variants of graph clustering are Hierarchical Clustering and Overlapping
Clustering. While there are some studies of both variants, they were not
previously considered simultaneously. The paper proposes a reasonable cost
function that combines both variants, and investigates algorithms that minimize
this cost function.

**Claims And Evidence:**

Claim 1: the proposed cost function for HOC makes sense.

The paper supports this claim by showing that it reduces to previous
variants for Hierarchical clustering trees.
It also discusses additional properties that supports the intuitive
meaning of this newly proposed cost function.

Claim 2: There are effective algorithms for minimizing the proposed cost
function.

The paper proposes such algorithms, and experimentally verifies their
performance.

**Essential References Not Discussed:**

N/A

**Experimental Designs Or Analyses:**

Experiments look fine. I also like the toy examples in the appendix.

**Methods And Evaluation Criteria:**

The main tool for the algorithm is local search.

**Other Comments Or Suggestions:**

It is an interesting result that may encourage others to look at both hierarchical and overlapping graph clustering models.

**Other Strengths And Weaknesses:**

N/A

**Questions For Authors:**

N/A

**Relation To Broader Scientific Literature:**

Yes, to the best of my knowledge.

**Theoretical Claims:**

Partially

---

> ### Author Rebuttal · Authors · 2025-04-01
>
> We thank the reviewer for the thorough review and the recognition of our work. We hope that our study will encourage other researchers to pay attention to hierarchical and overlapping graph clustering, since we think this hybrid structure has a great potential significance in recognizing real-world data organizations.
>
> ============================================
>
> (Sorry for occupying this space for the response to Reviewer 4xK8 due to the strict limit on character number.)
>
> Q7: In our proof of Th. 3.3, we focus on the ratio $cost^*_D/cost^*_P$ (Lemma B.4). We want to give an upper bound on this ratio such that proper bounds in terms of $n\cdot w(E)$ can be given to both of them ($cost^*_D$ is far from $n\cdot w(E)$, while $cost^*_P$ is far from 0, the farther the better). Since these two costs correspond to the same 2-OC partitioning, we consider the bilinear forms of both the primal and the dual versions of the cost, that is, bilinear combination of the total edge weights within clusters and the corresponding cluster sizes. Note that the total edge weights within a cluster is the multiplication of cluster size and density. When divided by a size term on both numerator and denominator, this ratio becomes related to density (although there is a size term multiplied on each density). Now the density of the densest subgraph can cover this ratio. Since density is closely related to node degree, we turn the density ratio to degree ratio for better comprehension. When the degree distribution varies wildly, the degree ratio $d max/d avg$ is not a good proxy of density ratio, and the bound in Th. 3.3 is not so good. But when the degree ratio is close to $1$, these two ratios get close, and now the cluster density of each cluster becomes uniform and the cost is mainly determined by the cluster sizes. Now the primal cost is roughly the sum of squares of two cluster sizes (ignoring the bad cut edges outside), while the dual cost is roughly twice of the product of two cluster sizes. This yields a high-quality upper bound for $cost^*_D/cost^*_P$.
>
> Q8: The stochastic block model (SBM) is not hierarchical. It has two variants, hierarchical SBM (HSBM) and overlapping SBM (OSBM), but there has been no variant like HOSBM yet, since an HOSBM, in our opinion, should be built on a proper formulation of HOC graph, just like what we propose in our submission. However, please note that any variant of SBM can be attributed to a (flat) stochastic adjacency matrix, in which the probability can be specifically set for each pair of nodes after calculation. In our settings, the hierarchical and overlapping features have been implied by $p_1, p_2, p_3$ that imply hierarchies and by $1-(1-p_3)^2$ that implies the density within overlapping areas (please refer to the 2nd paragraph of Appendix C.2).
>
> Q9: The formal definition of NMI for overlapping clustering has been given in Appendix C.3. It is a natural generalization of the classic NMI for non-overlapping partition.
>
> Q10: In Fig. 2, we have demonstrated the cost results for all synthetic datasets (see the 2nd column for both levels). For real datasets, since no baseline method can deal with the large graphs listed in Table 1 (with our computing environment of PC), we have no object to compare with. So we didn’t list the costs. However, the effectiveness of our algorithm has been evaluated in Fig. 1. Nevertheless, we have calculated the primal costs of the four networks in Table 1, those are $1.35E11$, $1.63E12$, $1.43E11$ and $1.65E10$ for Amazon, Youtube, DBLP-all and DBLP-cm, respectively.
>
> After all, regarding to W1 on the significance of Algorithm 1, please refer to our response to Reviewer 7SG8, the second item.
>
> Hope we have addressed all the concerns. If there are any remaining questions, we are happy to address them.

---

### Decision · Program_Chairs · 2025-05-01

**Decision:**

Accept (poster)

**Comment:**

Both of hierarchical and overlapping clustering are important topics in machine learning, and the current submission studies hierarchical overlapping clustering. All the reviewers gave positive evaluation on the submission, and it should be a clear accept for the paper.

The reviewers listed several downsides on the experimental studies of the paper, and I hope that the authors could take the reviewers' comments into account when preparing the next version of the paper.